# Borate driven heterogeneous networks for porous elastomers with improved tribological and mechanical performances

Yuhao Wu [1], Liguo Qin [1] ✉, Zeyu Ma [1], Mingqing Sun[1], Zheng Wang[1], Shan Lu[1], Xiaodong Huang[1], Wentao Xia[1], Hao Yang[1], Jianbo Liu[1], Ke Yan[1], Xin Ge[2] ✉, Sen Yang [3] ✉ & Guangneng Dong[1]

Porous polydimethylsiloxane (PDMS) elastomers produced using conventional phase separation methods often suffer from limited tribological and mechanical properties. In this study, we present a general approach for the fabrication of porous elastomers with borate-driven heterogeneous networks, which leverage the borate at the two-phase boundary to stabilize the phases, without the need for additives such as surfactants. Pure porous elastomers are created by removing the alcoholysis phases, with precisely controllable pore structure and mechanical performance. Interestingly, under saltwater lubrication, an atypical corrosion–lubrication phenomenon occurs on the porous elastomers, reducing the friction coefficient and wear rate by over 95 and 90%, respectively. Additionally, unique mechanical advantages, such as 1250% of stretch, were predictively imparted to the elastomers by various boric acid functional groups. Our method is generally applicable to various PDMS systems and offers a novel approach for designing pure porous materials for water-lubricated components and flexible sensors.

Double emulsions are formed via a phase separation, when two types of oils are sheared in specific proportions[1]. For example, beef soup exhibits oil-in-water phase separation between oil and water. Phase separation represents the transition from disorder to order in heterogeneous systems. To minimize free energy, colloids transform into two or more coexisting phases to achieve equilibrium[2]. The phase separation equilibrium is influenced by factors such as polymer–substrate interactions[3], temperature[4], pressure[5,6], and composition ratio[7,8]. In a system with a constant amount of two substances $a$ and $b$, the changes in the Gibbs free energies ($G_a$ and $G_b$) per mole are described as follows[9]:

$$\left(\frac{\partial G_a}{\partial T}\right)_P dT + \left(\frac{\partial G_a}{\partial P}\right)_T dP = \left(\frac{\partial G_b}{\partial T}\right)_P dT + \left(\frac{\partial G_b}{\partial P}\right)_T dP \quad (1)$$

where $T$ and $P$ represent the temperature and pressure of the system, respectively. By varying $T$, $P$, and the starting point of phase separation of components $a$ and $b$, a controlled phase separation can be achieved. This principle also allows for the regulation of the liquid–liquid phase separation in polymer networks. In colloid systems, phase separation and polymerization typically occur simultaneously, enabling the control of the network structures of the polymer via adjustments in temperature, pressure, and the polymer ratio.

For centuries, corrosion has been regarded as one of the factors that exacerbate friction and wear. However, metal friction components, such as bearings, blades, and axes, have to be used in a corrosive environment[10,11]. Friction can destroy the passivation film on the metal surface and promote corrosion[12]. To avoid metal corrosion, non-metallic materials such as poly(ether-ether-ketone), epoxy resin,

[1]Key Laboratory of Education Ministry for Modern Design and Rotor-Bearing System, Institute of Design Science and Basic Components, School of Mechanical Engineering, Xi'an Jiaotong University, Xi'an, PR China. [2]Department of Materials-Oriented Chemical Engineering, School of Chemical Engineering, Fuzhou University, Fuzhou, PR China. [3]School of Physics, MOE Key Laboratory for Nonequilibrium Synthesis and Modulation of Condensed Matter, Xi'an Jiaotong University, Xi'an, PR China. ✉e-mail: liguoqin@xjtu.edu.cn; gexin@fzu.edu.cn; yangsen@mail.xjtu.edu.cn

carbon composites, and polydimethylsiloxane (PDMS) have been used to replace metal in moving parts[13,14]. PDMS elastomers are critical base materials used in applications such as sensing[15-18], anti-icing[19-21], anti-fouling[22,23], and anti-wear[24]. Hydrophobic PDMS has a coefficient of friction (COF) of 0.8–1.2 when used with water-based lubricants, and the high COF limits the use of PDMS[25]. Porous PDMS can reduce wear because their pores can capture the wear debris, thus reducing wear. However, conventional methods for preparing porous elastomers, including emulsion templates[26,27], chemical foaming[28], and particle template methods[29], often face challenges in process control and compromise the frictional and mechanical performance of elastomers because of the numerous defects and impurities from surfactants and particle templates. An alternative approach involves fabricating porous elastomers using interpenetrating polymer networks (IPNs) by selectively degrading non-crosslinked subchains or extracting them from semi-IPNs or IPN precursors[30-32]. Existing studies on this method are limited and predominantly focus on composition ratio adjustments for pore structure control, ignoring the influence of pressure and temperature[33,34]. In addition to the PDMS network, another network in PDMS IPNs, such as poly(methyl methacrylate)[33,35], has significantly distinct chemical characteristics compared to those of PDMS, which may result in a significant decrease in the mechanical properties of the elastomer when removed. Therefore, limited research has been conducted on the use of this method to regulate the mechanical properties. However, manipulating the degree of phase separation, fabricating highly pure porous polymer elastomers, and regulating their tribological and mechanical properties are theoretically feasible[1,34-38].

This study presents a method for preparing heterogeneous-network porous PDMS (HNP-PDMS) elastomers, as shown in Fig. 1A. (For all abbreviations and their expansions, see Supplementary Table 1). Two networks were used: the first network consisted of D-PDMS with hydroxyl-terminated PDMS and boric acid (BA)[39-41], whereas the second network was derived from Sylgard 184 PDMS components (vinyl-terminated PDMS, silica-hydrogen bonded PDMS, platinum catalyst[42], and S-PDMS). The two networks, featuring similar main-chain properties, were driven by borate to achieve the heterogeneous phase separation. Prior to curing, the liquid mixture of Sylgard 184A + Sylgard 184B and dynamically crosslinked PDMS (S- and D-PDMS) was stirred to form multiple emulsions without any surfactants or solvents. Borates were incorporated into the PDMS network to stabilize the phase separation boundary to prepare the HNP-PDMS elastomers. HNP-PDMS@ boric acid (BA) could spontaneously form the corrosion-induced self-lubrication effect in saltwater (3.5% of NaCl solution), changing the lubrication state from boundary lubrication to mixed lubrication (Fig. 1B, C). The COF and wear rate were as low as approximately 0.05 and $2.72 \times 10^{-5}$ mm/N m, respectively. Mechanical performance advantages could be imparted to the porous elastomers, such as a stretch percentage of 1250%, by controlling the temperature, pressure, and borate sites (Fig. 1C). Using a wide variety of BA derivatives and silicone materials, this study offers a new approach for designing specialized porous elastomers for enhanced water lubrication performance and preparing flexible sensors.

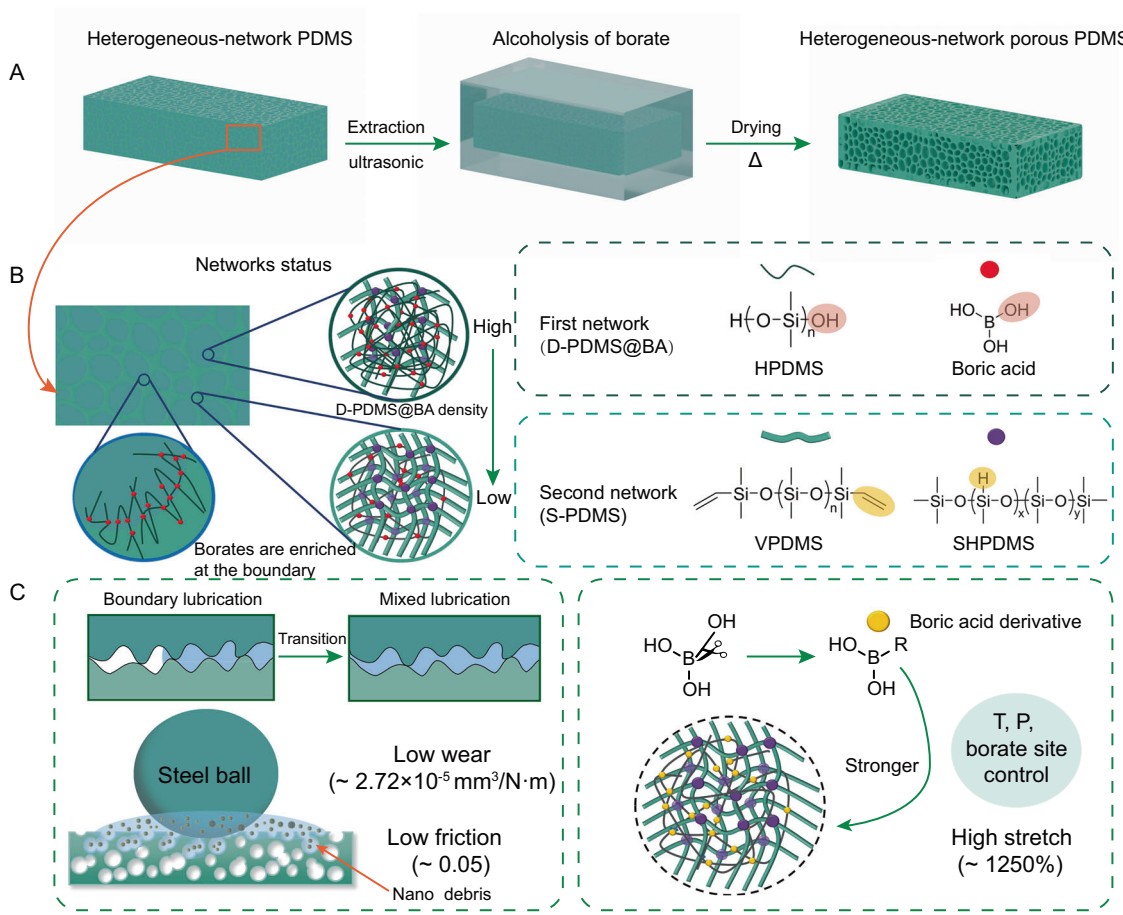

**Fig. 1 | Formation, tribological, and mechanical enhancements of HNP-PDMS. A** Preparation process of the HNP-PDMS elastomer. **B** Molecular chain entanglement in the heterogeneous network structure. The borates ensure the stability of the phase separation; in the spherical phase, a small amount of the second network is entangled, whereas in the porous skeleton, a small amount of the first network is entangled. **C** Schematic of tribological and mechanical enhancements.

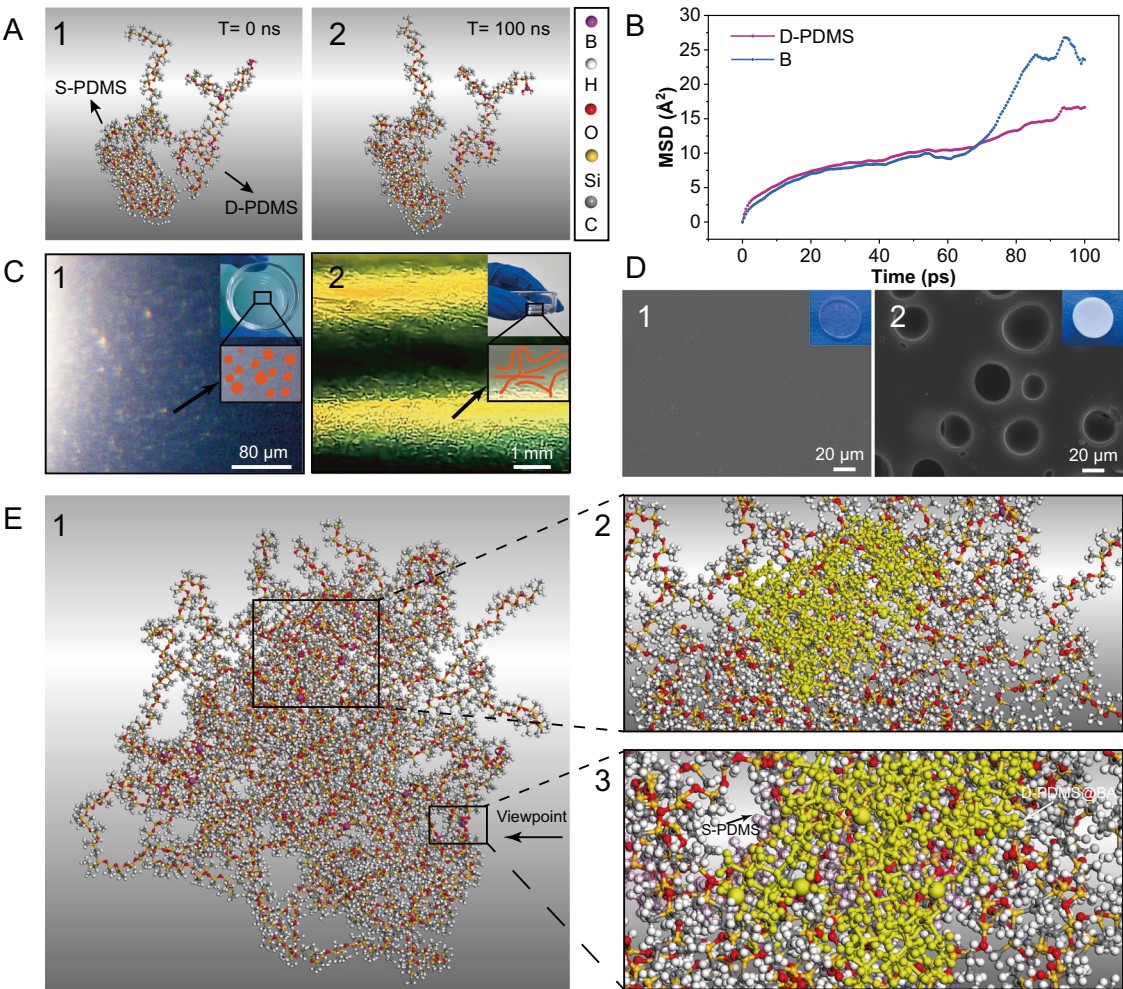

**Fig. 2 | Molecular mechanism of heterogeneous network formation and phase separation. A** Molecular dynamics (MD) simulation snapshots of the chains of a liquid mixture of Sylgard 184A + Sylgard 184B (S-PDMS) and dynamically cross-linked PDMS (D-PDMS) @boric acid (BA) at 0 and 100 ns. **B** Mean square displacement calculation results of the D-PDMS@BA chain and the boron atom. **C** Optical images after the expulsion of air bubbles under −0.1 MPa. **D** Scanning electron microscopy images of heterogeneous-network PDMS@BA and HNP-PDMS@BA cured at 125 °C and −0.1 MPa (the embedded images are optical images). **E** MD simulation snapshots of the mixed system of S-PDMS and D-PDMS@BA chains. The yellow chains represent D-PDMS@BA, and the purple chains represent S-PDMS.

## Results

### Phase separation

In this study, we examined the blending state of the mixed S-PDMS and D-PDMS@BA prepolymer liquid (with a mass fraction of D-PDMS@BA of approximately 20%). To investigate the effect of the interactions between the two polymer chains on phase separation, molecular dynamics (MD) simulations were performed. We simplified the two polymer networks and established four molecular models (Supplementary Fig. 1). Figure 2A presents MD simulation snapshots showing the mutually repulsive positions of the two polymer chains at 300 K, at 0 and 100 ns. The solubility parameters are listed in Supplementary Table 2; the difference in the solubility parameters between S-PDMS and D-PDMS@BA is greater than 0.5, indicating that S-PDMS and D-PDMS@BA are incompatible. Therefore, mixing the two phases require additional energy; otherwise, phase separation occurs. In the phase-separation mixture, as shown in Supplementary Fig. 2, all hydroxyl groups of H-PDMS reacted after curing to obtain HN-PDMS@BA. Fourier transform infrared spectroscopy (FTIR) results show that the borate group of D-PDMS@BA can be considered the distinguishing characteristic, compared to S-PDMS (for detailed discussion, see Supplementary Information, Section 2.1). Consequently, we considered the interaction between borate ester and silicone oil as

an important factor of incompatibility. The electrostatic potential of the repeating unit in D-PDMS@BA was calculated as shown in the Supplementary Fig. 3, to intuitively display the interactions of the borate bond with the siloxane backbone. The interactions suggest that the incompatibility is likely due to the repulsion between the B−O bonds and the oxygen in the Si−O backbone (for detailed discussion, see Supplementary Information, Section 2.1). Mean square displacement (MSD) results quantitatively showed that the system reached equilibrium at 95 ps (Fig. 2B). The boron atom exhibited a higher MSD than the D-PDMS@BA chain, which, combined with the electrostatic potential result, suggests that the borate increased the repulsion between the two chains. Consequently, we consider that increasing the D-PDMS@BA content (or the borate group content) intensifies the repulsion between D-PDMS@BA and S-PDMS, thereby enhancing the phase separation.

The initially polymerized D-PDMS@BA prepolymer was blended with the Sylgard 184 prepolymer and vacuumed in a drying oven to a pressure of −0.1 MPa, forming a transparent blend. As the pressure decreased, the solute precipitated and agglomerated through the spinodal decomposition mechanism, forming the microsphere and branched liquid phases (the phase separation is visible in Fig. 2C). According to the liquid−liquid phase separation mechanism[9], $\Delta G_{\text{mix}}$

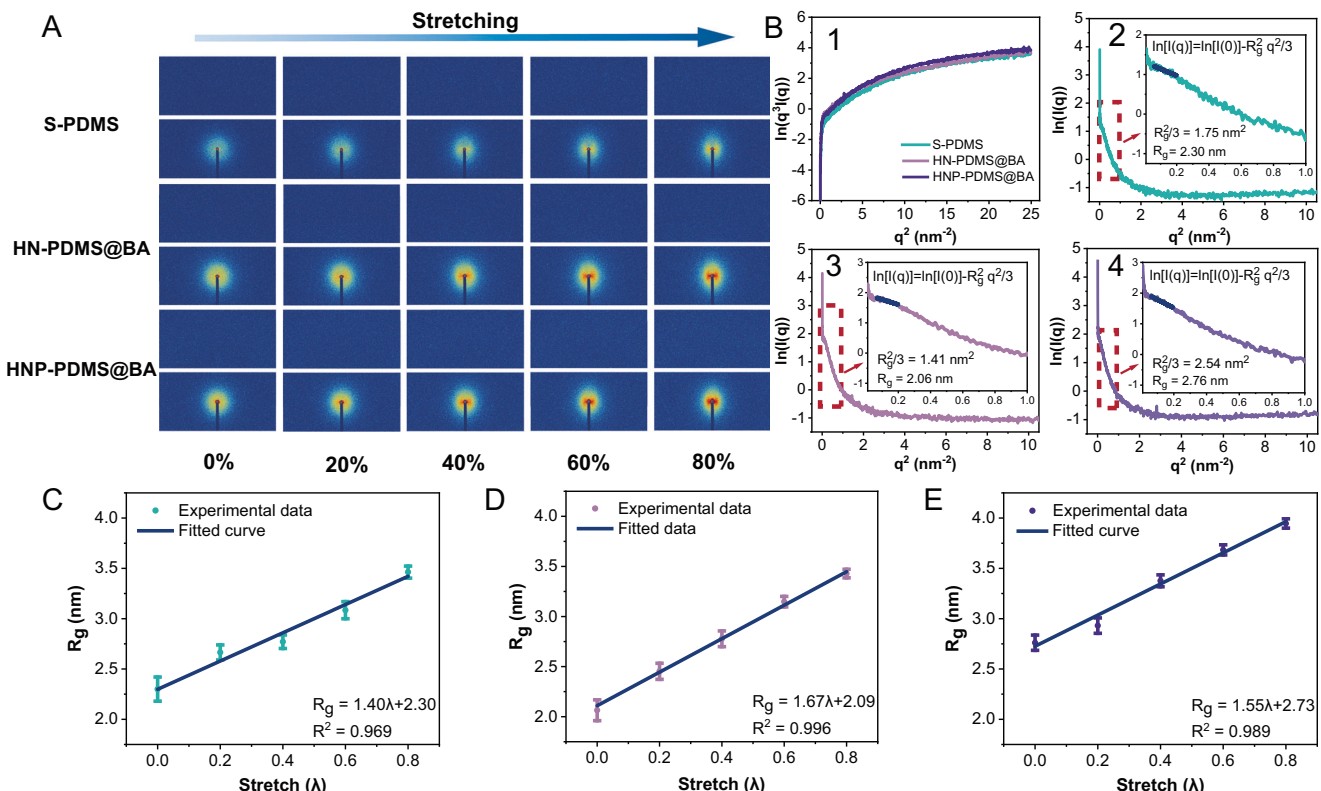

**Fig. 3 | SAXS analysis in situ stretching. A** 2D small-angle X-ray scattering patterns of S-PDMS, HN-PDMS@BA, and HNP-PDMS@BA under in situ stretching. The color gradient from blue to red indicates higher intensity levels. Each column corresponds to a different stretch level (0, 20, 40, 60, and 80%), and each row shows the scattering pattern for a specific polymer at the respective stretch levels. At 0% stretch, all three polymers exhibit symmetrical and concentrated scattering patterns. **B1** Porod curves for the various samples. **B2, B3, and B4** Guinier law analysis of small-angle X-ray scattering data for S-PDMS, HN-PDMS@BA, and HNP-PDMS@BA. Gyration radius ($R_g$) as a function of the stretching ratio for **C** S-PDMS, **D** HN-PDMS@BA, and **E** HNP-PDMS@BA. The intercepts are the $R_g$ of the elastomer at 0% stretching after fitting. The error bars show the standard deviation of the mean, computed from the data of $qR_g < 1$.

remains above 0 at air pressures of −0.1 and 0.0 MPa. The two phases are surrounded by borates, as shown in Fig. 1B. To better understand the molecular interactions involved in this process, additional MD simulations were performed with a mixed model of nine S-PDMS and nine D-PDMS@BA chains (Fig. 2E). During system relaxation, agglomeration of the D-PDMS@BA chains and entanglement of the S-PDMS chains occurred (Fig. 2E2, E3). A radial distribution function was used to study the aggregation of boron atoms (Supplementary Fig. 4). The appearance of a wide peak indicated that the boron atoms agglomerated within the 3–18 Å range (Fig. 2E2). The aggregation of boron atoms was hindered by the Si−O skeleton of S-PDMS or D-PDMS@BA during the agglomeration process, leading to the entanglement or encapsulation of D-PDMS@BA and S-PDMS (Supplementary Fig. 3 and Fig. 2E3). Because of the aforementioned state of the mixture, the prepolymer could be cured to obtain a heterogeneous network of PDMS with a smooth surface (Fig. 2D1). When the unstable D-PDMS@BA network was removed from HN-PDMS@BA using ethanol, owing to the alcoholysis of borate (Supplementary Fig. 5), an HNP-PDMS elastomer was obtained (Fig. 2D2). S-PDMS was encapsulated by a small amount of D-PDMS@BA, forming the polymer framework, and the reactions shown in Supplementary Fig. 6, facilitated this state (further discussion is provided in Supplementary Information, Section 2.1). The D-PDMS@BA phase was also surrounded by a small number of S-PDMS chains.

## Small-angle X-ray scattering analyses
To understand the molecular orientations, phase structure, and microregions within the elastomers, small-angle X-ray scattering (SAXS) analyses were performed. The 2D SAXS patterns for S-PDMS,

HN-PDMS@BA, and HNP-PDMS@BA after in situ stretching are shown in Fig. 3A, which illustrates the arrangement of the molecular chains. First, the SAXS results for elastomers at 0% stretch are discussed. As depicted in Supplementary Fig. 7, all samples displayed a characteristic peak at scattering vector $q \approx 0.08$ nm$^{-1}$, corresponding to a calculated periodicity of 78.5 nm for the microphase-separated domains. Porod curves for S-PDMS, HN-PDMS@BA, and HNP-PDMS@BA exhibited a positive deviation in the high-angle area (Fig. 3B1), suggesting the presence of a diffuse transition layer at the two-phase interface rather than an abrupt change in electron density. This phenomenon was attributed to the entanglement of the Si−O chains at the interface (Fig. 1B). Free volume fluctuations were observed within each phase of the soft−hard region, with no clear interface, suggesting the formation of a complex spatial distribution characterized by mutual wrapping and entangling[43–45]. Moreover, the elastomers displayed an initial parabolic peak in the low-angle region, followed by an increase (Supplementary Fig. 8), representing partial folding of the molecular chain in a spherical-random coil configuration[46]. Figures 3B2 – 4 show the Guinier curves of the natural logarithm of the scattered intensity ($\ln(I(q))$) versus $q$. All Guinier curves displayed concave curved lines with rectilinear segments, indicating that the elastomers belonged to a spherical polydisperse system. The radius of gyration ($R_g$) of a scatter, which generally remains constant, quantitatively represents the molecular chain[47,48]. When $qR_g < 1$, $R_g$ is determined by calculating the slope in the low-angle region. With the addition of D-PDMS@BA, the $R_g$ decreased, indicating a shorter molecular chain length for D-PDMS@BA[49]. When D-PDMS@BA was removed using ethanol, the $R_g$ of HNP-PDMS@BA exceeded that of S-PDMS, suggesting that the removed D-PDMS@BA network was entangled with short S-PDMS

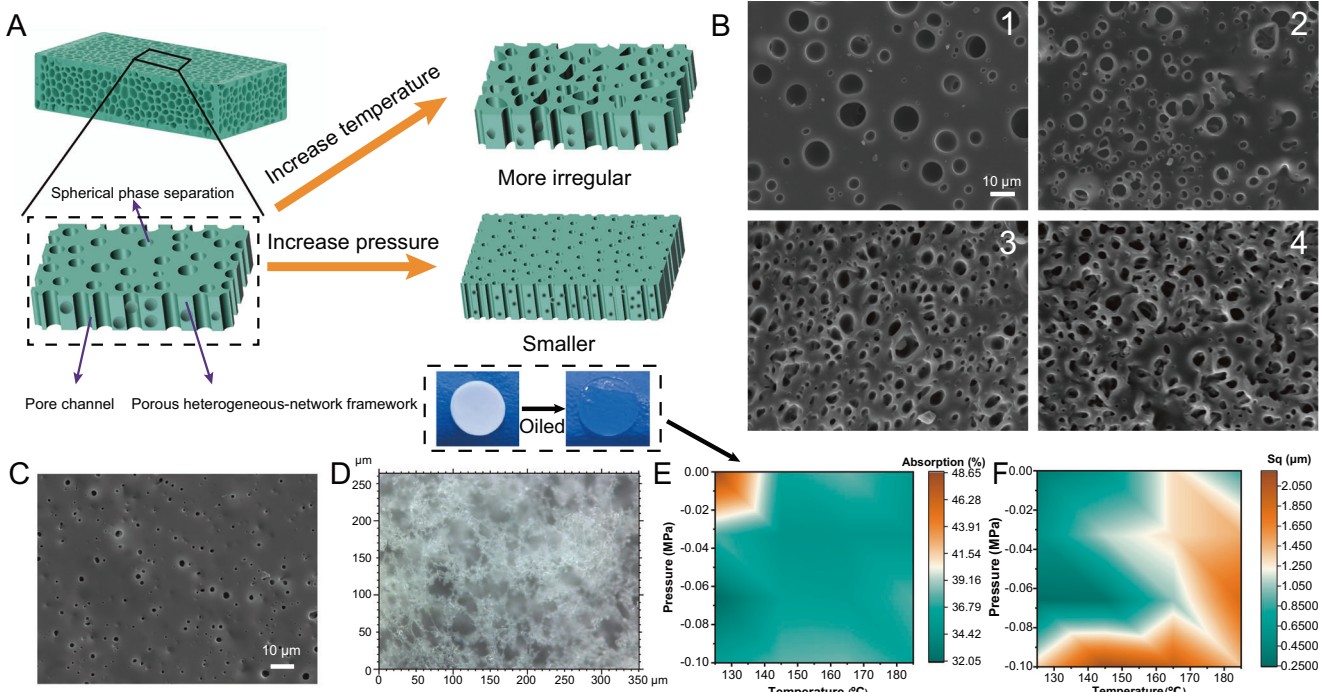

**Fig. 4 | Morphology and porous structure of HNP-PDMS@BA. A** Schematic illustrating the control of surface pore morphology. The inset optical images show HNP-PDMS@BA before and after oil absorption. **B** Pore morphology under pressures of −0.1, −0.066, −0.033, and 0 MPa at 125 °C. **C** Pore morphology under a pressure of 0 MPa at 125 °C. **D** Passage and structure in the lower layer of the porous surface. Changes in **E** oil absorption and **F** root mean square (Sq) roughness as a function of temperature and pressure.

molecular chains. These findings aligned with our hypothesis regarding the multiple emulsion states of the HNP-PDMS@BA prepolymer. Based on the MD simulation results, phase separation analysis, SAXS results, and scanning electron microscopy (SEM) image shown in Fig. 2C, we inferred that the HN-PDMS possessed a heterogeneous network structure with interpenetrating networks at the phase boundaries (Fig. 1B). HNP-PDMS@BA featured not only a porous structure but also longer molecular chains compared to those of S-PDMS and HN-PDMS@BA.

We stretched the elastomers from 0 to 80% and observed the SAXS patterns during this process (Fig. 3A). As the stretch increased, the isotropy was disturbed, and the molecular chains became oriented toward the direction of stretching. Among the three kinds of samples (S-PDMS, HN-PDMS@BA, and HNP-PDMS@BA), the anisotropy of HNP-PDMS@BA was most pronounced. Studies have shown that the tensile strength and stiffness of polymers improve significantly when their molecular chains align along the stretching direction[50], as this arrangement allows the chains to better resist the external stretching forces. The $R_g$ values were fitted to a straight line, as shown in Fig. 3C – E, revealing a linear correlation between the radius of rotation and the stretching ratio during the stretching process. The introduction of the D-PDMS@BA network improved the linear fitting ($R^2 = 0.996$), indicating that the D-PDMS@BA network promoted the alignment of the network toward the direction of the force during stretching. Notably, the slope values of the fitted lines for S- and HNP-PDMS are numerically similar to the tensile modulus (as discussed in Supplementary Information, Section 2.4), demonstrating that the slope values represent the strength and stiffness of the molecular chain.

## Morphology and porous structure induced by network sacrificing

The metastable state of the phase separation varied with increasing temperature and pressure (Fig. 4A). The diffusion coefficients of the two polymers changed depending on the temperature and pressure.

MSD analysis was used to investigate the diffusion process of D-PDMS@BA, revealing that higher temperatures accelerated its diffusion, whereas the opposite was true for pressure (Supplementary Fig. 9 and Table 3). Consequently, by controlling the temperature and pressure, the entanglement and separation of the two chains can be regulated to achieve different network structures. Subsequently, the unstable D-PDMS@BA network in HN-PDMS@BA formed elastomers with diverse pore structures after undergoing alcoholysis. Supplementary Fig. 10 illustrates the mass change ($\triangle m$) from HN-PDMS@BA to HNP-PDMS@BA as the curing temperature increases, showing that the reduction in mass diminishes with higher curing temperatures. The mass of dissolved D-PDMS@BA in S-PDMS was significantly lower than the mass of the first network (20 mg), suggesting that some D-PDMS@BA remained entangled with S-PDMS (further discussion is provided in Supplementary Information, Section 2.2).

Figure 4B shows the surface morphology of HNP-PDMS@BA elastomers synthesized at various temperatures. Fractal dimensions and pore roundness (θ) were quantified to investigate the phase separation using ImageJ software. As the temperature increased, the pores appeared denser (Supplementary Figs. 11 and 12), and θ progressively decreased (Supplementary Fig. 13), indicating that the pores became more irregular with increasing temperature. This phenomenon was attributed to the accelerated curing speed at higher temperatures, which disrupted the semi-stable state of the D-PDMS@BA sphere phase and drove the D-PDMS@BA chains to migrate toward the S-PDMS chains[9]. Conversely, when the pressure was improved (−0.1–0 MPa), the pores became smaller (5.16–1.16 μm) and sparser (Fig. 4C and Supplementary Fig. 14). This was because a higher pressure compressed D-PDMS@BA, making it difficult to diffuse, leading to a transition to a semi-stable state and smaller pores. These findings aligned with the results of our MD simulations. Supplementary Fig. 15 shows the SEM images of the different surfaces. Although some surfaces appeared smooth at high pressures and high temperatures, sparse pores were still seen in the SEM images (Supplementary

Fig. 16A). The surfaces exhibited a ridge-like topography in three dimensions, at high pressure and high temperature, as seen in the confocal laser scanning microscopy images (Supplementary Fig. 17). As the temperature increased, the pores on the surfaces tended to shrink, eventually disappearing at 0 MPa. Under the same coordinate axis scale (objective lens 50×), the cross-sectional outline of HNP-PDMS@BA at 0 MPa approximated a straight line (Supplementary Fig. 18). The surface roughnesses of HNP-PDMS@BA elastomers, synthesized at various temperatures as shown in Fig. 4F, show a clear demarcation, providing a reference for designing friction surfaces. To investigate the oil-absorption capacity of the porous elastomers, methyl silicone oil was injected into HNP-PDMS@BA elastomer samples. An absorption map was obtained by calculating the porosity of HNP-PDMS@BA under 16 different conditions (Fig. 4E and Supplementary Eq. 12). The absorption percentage of HNP-PDMS@BA remained stable between 30 and 50% when the temperature and pressure were varied.

Subsequently, to investigate the internal microscopic pore structures of the porous elastomers, the internal morphology of the phases in HNP-PDMS@BA was observed using confocal laser scanning microscopy because of its transparency (Fig. 4D), revealing an internal interconnected and interpenetrating structure. For further investigations of the internal framework, the porous elastomer was cut diagonally and observed using SEM (Supplementary Fig. 16B). When the HNP-PDMS@BA elastomer, filled with silicone oil, was cut, the oil flowed out of the pores within 30 s (Supplementary Fig. 19). To investigate the smaller-scale features, the nanochannels in the elastomer were analyzed using transmission electron microscopy. These nanochannels, resembling cylindrical pipes with diameters less than 500 nm, extended through the elastomer (Supplementary Fig. 20). Additional pore details are shown in Supplementary Fig. 21. At an air pressure of −0.1 MPa, HNP-PDMS@BA primarily formed micron-sized pores (>0.5 μm) and mesopores (<50 nm), while at a pressure of 0.0 MPa, it formed more macropores (>50 nm). The absorption of silicone oil (30–50%) was significantly higher than that of mercury (porosity of approximately 10%), which may be attributed to the swelling of HNP-PDMS@BA owing to oil absorption[23,51]. These findings suggested that HNP-PDMS@BA featured a microsphere airspace and interconnected micro-nano channels. To compare the pore structures of elastomers prepared using our method with those generated using conventional methods, we prepared porous samples with the same mass percentage as the first network. Supplementary Fig. 22 shows the SEM images of the surface morphology and cross-section of porous PDMS. The cross-sections of all samples were porous, and their surfaces were nonporous. The pores in the samples prepared by the particle template method were larger than those in the samples prepared by our emulsion method; larger pores were likely to lead to limited elongation at break and reduced toughness. This finding implied that our method not only imparted a unique porous structure to the elastomers (with adjustable surface pores and interconnected internal pores) but also laid the foundation for excellent tribological properties (as discussed in the next Section).

## Friction performance

To investigate the ultra-low friction and wear behavior of HNP-PDMS@BA, we conducted ball-plate experiments under saltwater lubrication (Fig. 5A). In this section, P1 and P2 represent HNP-PDMS@BA-125/−0.1 MPa, and HNP-PDMS@BA-125/0 MPa, respectively, while G and S represent the GCr15 and $Si_3N_4$ balls, respectively. The morphology and wettability of the surface strongly affect the COF. In particular, if the surface wettability is improved and combined with appropriate surface structure, a transition of the lubrication regime can lead to a decrease in the COF[52,53]. We started the friction investigations by exploring the differences between the lubricating films on hydrophobic and hydrophilic surfaces during friction. Because the

contact angle (CA) of saltwater is similar to that of pure water, pure water was used to simulate the wetting state of saltwater[54]. CAs for S-PDMS, P1, P1 after one friction test and P1 after three consecutive friction tests were measured as approximately 105, 101, 73, and 77°, respectively (Supplementary Fig. 23A–D). In the MD simulations, the CAs of the surfaces were set such that the error in each case was within 3°, compared to the experimental CAs (see the "Methods" section for the details). MD simulations were performed to investigate the differences in the water film on hydrophobic and hydrophilic surfaces (Supplementary Fig. 24 and Movies 1, 2). Results of the interactions between the ball, water, and the hydrophobic and hydrophilic surfaces at 10 ns are shown in Fig. 5B, and Supplementary Fig. 25. Compared to the hydrophobic surface, a complete water film was formed between the ball and the hydrophilic surface. Correspondingly, the water density distribution is shown in Fig. 5C. It shows that when only the wetting effect is considered, hydrophilic surfaces are more likely to cause water to accumulate at the bottom of the ball than hydrophobic surfaces. This hydrophilic shift is likely to lead to a COF close to that of previous studies[25,55]. Water was easily absorbed in the lower part of the ball and the water film density on the hydrophilic surface was more than twice that of the hydrophobic surface (Fig. 5C). As shown in Fig. 5D, the COF of P1-G and P2-G continually decreases sharply during the friction process and finally stabilizes. After stabilization, the average COF of S-PDMS-G, P1-G, P2-G, P1-S, and P2-S was 1.06, 0.17, 0.05, 1.17, and 1.13, respectively (Fig. 5E). By comparison, COF reduced by over 95% between S-PDMS-G/P1-Si/P2-Si and P2-G. As shown in Fig. 5F, P2-G wear volumes and rates reduced by over 90% between S-PDMS-G/P1-Si/P2-Si and P2-G. After friction, S-PDMS showed clear furrow wear, and P1 displayed nearly no wear marks (Fig. 5G, H). The other worn surfaces also showed fewer wear marks than S-PDMS, and the capture of wear debris by pores was observed on all surfaces (Supplementary Figs. 26 and 27).

When GCr15 was used as the friction pair, electrochemical corrosion continued to occur (Supplementary Fig. 28). We inferred that this electrochemical corrosion was beneficial to reduce the COF and wear, because the COF and wear rate were higher when $Si_3N_4$ was used as the friction pair. In the corrosion process, wear debris consisting of nanoclusters was produced in saltwater, which enhanced water adsorption under the influence of iron oxide (Supplementary Fig. 29). Meanwhile, the wear debris was captured by the pores (Supplementary Fig. 26), leading to the surface wetting change from hydrophobic to hydrophilic (Supplementary Fig. 23B, C), which in turn thickened the lubrication film. With the hydrophilic effect, pores store water easily, causing local dynamic pressure effects near the pores[56]. The Sq of P1 with large pores was 3.8 times higher than that of P2, as shown in Fig. 4F, increasing the initial wear and the wear debris capture capability (Supplementary Fig. 26D, E), which in turn thickened the lubrication film. Therefore, P1 exhibited faster friction reduction and a lower wear rate (Fig. 4B and D). Overall, when GCr15 and HNP-PDMS@BA (P1 and P2) were rubbed against each other, corrosion occurred and hydrophilic debris was produced, which led to hydrophilic HNP-PDMS@BA surfaces. The coordinated effect of the hydrophilic surface and the porous structure with stored liquid reduced abrasive and three-body wear, causing the boundary lubrication to transform into mixed lubrication, which in turn resulted in ultra-low friction and wear (Fig. 5I). The transformation is reflected by the shear stress and COF under a single reciprocating cycle (Supplementary Fig. 30). Compared to the unreduced state (Supplementary Fig. 30A–D), a clear sliding platform and a stable low COF appeared in the COF-reduced cycle (Supplementary Fig. 30E, F). At higher loads (4 N and 6 N), the sliding plateau disappeared, indicating that the friction was dominated by the elastic deformation of the surface[57]; however, the average COF continued to be low (Supplementary Fig. 31). Furthermore, the ability to improve the tribological properties was highly stable, as shown in Supplementary Figs. 32–39. P1 exhibited

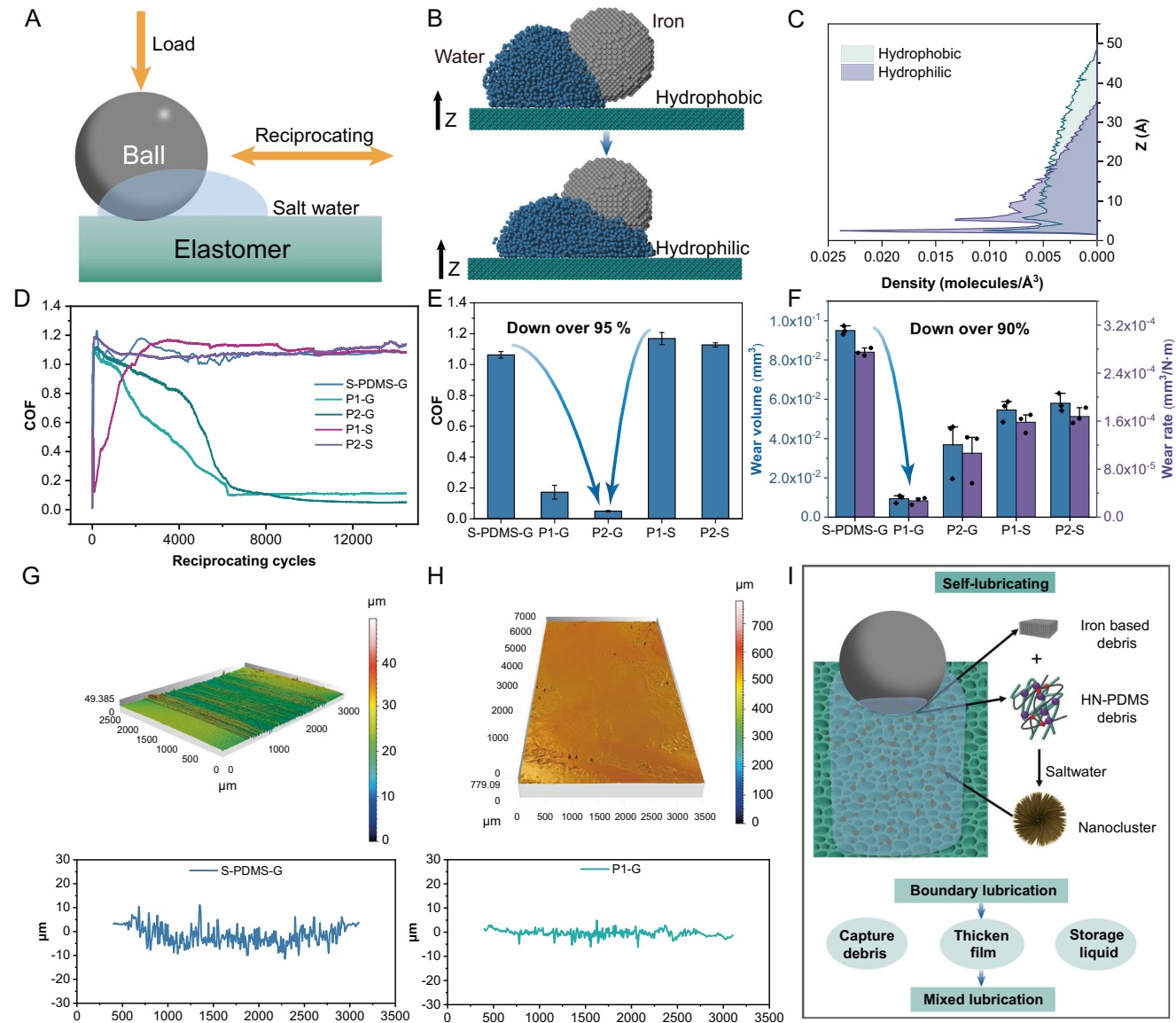

**Fig. 5 | Tribological behavior of HNP-PDMS@BA. A** Schematic of the sliding configuration. Molecular dynamics simulation results: **B** water interacting with a steel ball and a hydrophobic or hydrophilic surface at 10 ns (blue particles are the water molecules, and gray particles are the iron atoms; the cyan surface simulates the hydrophobic or hydrophilic surface) and **C** water density map in the Z direction. **D** The COF–cycle curve of different friction samples at 2 N and 4 Hz. **E** Average COF at 2 N and 4 Hz after stabilization. The error bars show the standard deviation of the mean, computed from the COF at the steady cycle state. **F** The wear volume and rate of different friction samples. The error bars show the standard error of the mean, computed from three samples. The 3D profile and 2D cross-section of **G** S-PDMS and **H** P1 after 14,400 reciprocating cycles. **I** Schematic of corrosion-induced self-lubrication.

low friction and wear even after cyclic tensile, friction fatigue tests, or under heat and UV radiation (Supplementary Information, Section 2.3, for a detailed discussion).

## Mechanical performance

The degree of macromolecular mixing and phase separation depended on the solubility of Sylgard 184A, Sylgard 184B, S-PDMS in D-PDMS, and the ability of D-PDMS to swell with Sylgard 184A and Sylgard 184B[32]. Based on our MD simulations and experiments, we inferred that the immiscibility of S-PDMS and D-PDMS prepolymers was closely related to the presence of borate. Consequently, we precisely controlled the mass of D-PDMS@BA to achieve different phase separations, resulting in elastomers with varying mechanical performances. As the proportion of S-PDMS was incrementally increased in D-PDMS, and when the mass percentage of D-PDMS was less than 20%, SEM analysis revealed the disappearance of the porous structure

(Supplementary Fig. 40). This result was attributed to the complete dissolution of the D-PDMS@BA solute by the S-PDMS solvent, leading to a transition to a homogeneous network polymer. By investigating the changes in the mechanical performance during the transition, we determined that a lower amount of D-PDMS@BA led to higher toughness and breaking strength in HN-PDMS (Supplementary Fig. 41A; details are presented in Supplemental Discussion 2.4). This result implied that a porous elastomer with minimal D-PDMS could improve the mechanical performance. The tensile modulus of the HNP-PDMS@BA elastomer improved to nearly 1.6 MPa (Supplementary Fig. 41D), and was consistent with the $R_g$ results from SAXS analyses. Generally, the compression modulus of PDMS is similar to its tensile modulus, and neither moduli are influenced by the curing pressure (Supplementary Fig. 42). However, the compression modulus of HNP-PDMS@BA could be adjusted from 0.25 to 1.25 MPa by changing the curing temperature and air pressure (Fig. 6A, B); the

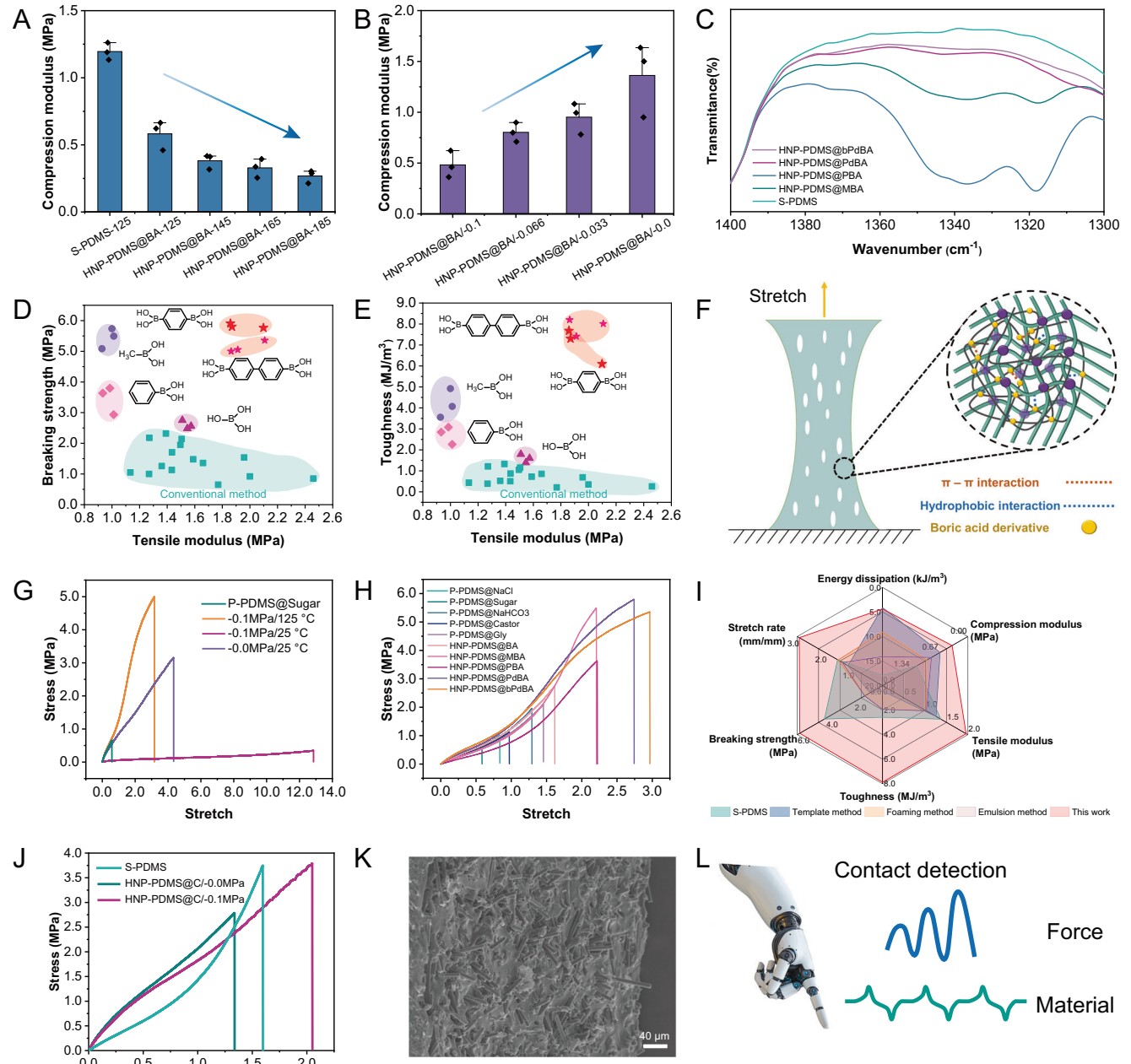

**Fig. 6 | Mechanical properties and potential applications of the HNP-PDMS series.** Compression modulus as a function of **A** temperature (fixed pressure: −0.1 MPa) and **B** pressure (fixed temperature: 125 °C). The error bars show the standard error of the mean, computed from three samples. **C** Fourier transform infrared spectra of HNP- and S-PDMS. Comparison of the **D** breaking strength and **E** toughness for different HNP- and porous PDMS elastomers prepared via conventional methods. The molecular structures represent the boric acid derivatives used in the different HNP-PDMS elastomers. **F** Molecular chain interaction diagram when a benzene ring is introduced. **G** Comparison of the stress–stretch curves of HNP-PDMS@bPdBA, cured under varying pressures and temperatures, and P-PDMS@Sugar. **H** Comparison of the stress–stretch curves for different elastomers. **I** Comparison of six critical parameters for different elastomers. **J** Comparison of mechanical performance of HNP-PDMS@C and S-PDMS. **K** Dispersion of carbon fibers within the through-pores of HNP-PDMS@C. **L** Application example of HNP-PDMS@C.

compression modulus negatively correlated with temperature and positively correlated with pressure. Overall, the results demonstrated the basic properties of a heterogeneous elastomer network when BA was used as the crosslinker and that the heterogeneous network was highly controllable and versatile.

**Improved mechanical properties and potential applications**
Generally, pores tend to reduce the breaking elongation and modulus[58]. In this section, we demonstrate that this limitation can be avoided by introducing BA derivatives within a heterogeneous

polymer network. The following networks were used: D-PDMS@methylboric acid (MBA), D-PDMS@phenylboric acid (PBA), D-PDMS@phenyldiboric acid (PdBA), and D-PDMS@biphenyl diboric acid (bPdBA). FTIR results confirmed the presence of D-PDMS in the new HNP-PDMS elastomers[59] (Fig. 6C). Next, we evaluated the mechanical properties. As shown in Fig. 6D and E, the toughness and breaking strength significantly improved in the new HNP-PDMS elastomers. D-PDMS, D-PDMS@MBA, and D-PDMS@PBA exhibited a reduced tensile modulus, whereas D-PDMS@PdBA and D-PDMS@bPdBA showed an increased tensile modulus. Evidently, the

introduction of the methyl and benzene rings enhanced the toughness of the HNP-PDMS elastomers. This improvement was attributed to the hydrophobic and $\pi$–$\pi$ interactions of the benzene rings in the continuous phase of the heterogeneous polymer networks (Fig. 6F). Increasing the borate content also increased the tensile modulus of the porous elastomers. Therefore, the HNP-PDMS@PdBA and HNP-PDMS@bPdBA elastomers exhibited high tensile moduli (approximately 2 MPa), breaking strength (approximately 5.4 MPa), and toughness (approximately 8 MJ/m³) (Fig. 5E). Similarly, HNP-PDMS@bPdBA demonstrated controllability of the tensile modulus in the 0.15–2 MPa range, by adjusting the curing pressure and temperature (Fig. 6G and Supplementary Fig. 43B). Surprisingly, the stretch percentage could be improved to 1250%, which is attributed to the increase in the degree of viscous deformation of HNP-PDMS@bPdBA, as shown in Supplementary Fig. 44 (see Supplementary Information, Section 2.5, for more details). Additionally, the performance of HNP-PDMS@bPdBA was better than that of P-PDMS@Sugar (Fig. 6G). Subsequently, using six key parameters, we compared the mechanical performance of the HNP-PDMS elastomers with those of other methods and S-PDMS (Fig. 6H, I). The results highlighted the overall superior performance of HNP-PDMS elastomers, particularly stretch rate, toughness, and breaking strength. Theoretically, we could modify the continuous phase, introduce arbitrary functional groups, and endow porous elastomers with unique properties.

To validate our method on silicone-based materials, porous Ecoflex silicone@BA elastomers were prepared using commercial Ecoflex silicone with a composition similar to that of S-PDMS[60]. We could successfully apply our method to prepare the new porous silicone structures (Supplementary Fig. 45A, B). The breaking strength and toughness were significantly improved (Supplementary Fig. 45C, D). The introduction of D-PDMS@BA significantly delayed the fracture of the elastomer, and the toughness increased to 150%, despite the elastomer being porous. Furthermore, we successfully prepared new porous elastomers using three other PDMS materials (DC170, DC527, and RTV615), as shown in Supplementary Fig. 46. These porous elastomers also exhibited elongation improvement potential. These results proved that our method was applicable to the preparation of mechanically enhanced porous elastomers for all PDMS systems.

PDMS is inherently insulating; therefore, to prepare flexible sensors, conductive properties need to be introduced to PDMS materials. We used carbon fibers with a diameter of 7 μm and a length of 33.6 μm as the conductive material (Supplementary Fig. 47). To achieve conductivity in PDMS/silicone materials, rather than trying to form pathways by directly mixing PDMS with conductive materials (Supplementary Fig. 48A), the conductive material is usually connected within the polymer using a preparation method similar to that for preparing porous materials[61,62]. Therefore, based on the formation mechanism of HNP-PDMS@BA, we added carbon fibers to PDMS and successfully prepared conductive PDMS (HNP-PDMS@C, resistance <1 kΩ), without the need for additional surfactants or emulsions (Supplementary Fig. 48B, C). The mechanical performance of HNP-PDMS@C/−0.1 MPa is better than that of S-PDMS (Fig. 6J), likely owing to the dispersion of carbon fibers within the through-pores of the HNP-PDMS (Fig. 6K). This flexible conductive material can be used as the skin of bionic robot hands, for example, to sense force and identify materials (Fig. 6L). To demonstrate this application effect, HNP-PDMS@C/−0.1 MPa was encapsulated in S-PDMS, bent, and fixed on a robotic finger for contact detection (Supplementary Fig. 49A). The material acted as a strain and triboelectric sensor. As the contact force increased from 1 to 4 N, the change rate of resistance increased accordingly (Supplementary Fig. 49B). According to the peak value and width of the wave[41], various materials can be distinguished as shown in Supplementary Fig. 49C, demonstrating practical potential applications. Overall, our method enables the emulsion-free formation of high-performance porous elastomers and flexible sensing materials.

## Discussion

We investigated the formation process of borate-driven heterogeneous elastomer networks and proposed a general method for the design of specialized porous elastomers that demonstrated excellent tribological and mechanical performances. When the mass percentage of D-PDMS in the prepolymer emulsion exceeded 20%, the emulsion disintegrated under shear, resulting in a double emulsion, without requiring the use of any surfactant or additive. Based on the heterogeneous network structure and pure composition, unique properties can be imparted to the porous elastomers using different BA derivatives (the hundreds of BA derivatives that are readily available). Although we tested only a limited number of BA derivatives, our approach demonstrated widespread applicability.

In this study, the HNP-PDMS elastomers that were prepared and tested exhibited atypical ultra-low friction and wear behavior under corrosion, which deviated from the general belief that corrosion aggravates wear. The toughness of the HNP-PDMS elastomers was nearly 16 times higher than that obtained using the conventional particle template method, when hydrophobic groups were introduced. The pore structure and the degree of continuous-phase network crosslinking were regulated by varying the temperature, pressure and the crosslinking site, which in turn altered the frictional and mechanical performances of the porous elastomer. Flexible contact sensing was achieved by simply incorporating carbon fibers into the borate-driven heterogeneous networks of PDMS elastomer. Our approach enabled the production of highly pure and controllable porous materials, with improved tribological and mechanical performances, and is generally applicable to all PDMS systems. Our study can bring new ideas to the design of water-lubricated components and porous flexible sensors.

## Methods
### Materials
HPDMS (average Mn ~550) was sourced from Sigma-Aldrich. A PDMS prepolymer (Sylgard 184, Sylgard 170, and Sylgard 527) and methyl silicone oil (100cs) were purchased from Dow Corning (USA). Ecoflex 00–31 and RTV615 were obtained from Smooth-On (USA) and Momentive (USA). BA, MBA, PBA, PdBA, and bPdBA were obtained from Shanghai Macklin Biochemical Technology Co. Ltd. Ethanol ($C_2H_6O$), NaCl, and cane sugar salts were purchased from Tianjin Fuyu Fine Chemical Co. Ltd. Castor oil and glycerin were obtained from Sinopharm Chemical Reagent Co. Ltd. and Shanghai Acmec Biochemical Technology Co. Ltd, respectively. Baking soda was purchased from Tongbai Boyuan New Chemical Co. Ltd. Ecoflex0031 silicone was obtained from Nantong Jingwei Biotechnology Co. All chemicals were used as received without further purification, and deionized water was used for all experiments and tests.

### Fabrication of dynamically crosslinked polydimethylsiloxane
BA, MBA, PBA, PdBA, and bPdBA were dissolved in absolute ethanol and stirred. Subsequently, HPDMS and a BA or BA derivative solution were mixed in a vial, maintaining the molar ratio of Si−OH to B−OH at 1:1. A mixture of diol, BA, or BA derivative and ethanol was heated to 70 °C and stirred for 2 h in a closed environment to form the oligomeric products. Then, the reaction mixture was heated at 100 °C in an open environment to remove ethanol and water, further promoting the reaction to obtain D-PDMS.

### Fabrication of heterogeneous-network porous polydimethylsiloxane elastomer
Sylgard 184 A and Sylgard 184B were mixed in a mass ratio of 10:1. D-PDMS was added to the Sylgard 184 mixture at mass percentages of 8, 10, 11, 13, 16, and 20%. The mixture was stirred for 30 min, which in turn removed any bubbles by vacuuming. To facilitate the formation of the heterogeneous network, a final cure of HN-PDMS was performed at

approximately 25, 125, 145, 165, or 185 °C under pressures of 0, −0.033, −0.066, or −0.1 MPa. Finally, the unstable D-PDMS network was removed by ultrasonic cleaning of the HN-PDMS elastomer with ethanol to obtain the HNP-PDMS elastomer.

### Fabrication of control elastomers

S-PDMS was prepared by mixing Sylgard 184A and Sylgard 184B in a 10:1 ratio and curing the mixture. The sugar template, salt template, castor oil emulsion template, glycerin emulsion template, and baking soda foam were added to the Sylgard 184 mixture at a mass percentage of 20%. The mixture was stirred for 30 min, which in turn removed any bubbles by vacuuming. Then, the mixture was cured at 125 °C under −0.1 MPa to obtain porous PDMS. The sugar, salt, and baking soda foam templates were removed using deionized water. The castor oil emulsion template and glycerin emulsion were removed using ethanol. Ecoflex 0031 and porous Ecoflex silicone@BA were cured at 25 °C under atmospheric pressure for 12 h. The other preparation processes were the same as those used to prepare HNP-PDMS. The preparation processes of DC170@BA, DC527@BA, and RTV615@BA were the same as HNP-PDMS.

### Fabrication of flexible sensor material

The preparation method of HNP-PDMS@C differs from that of HNP-PDMS@BA; the carbon fiber is added (25% of the total mass) during the mixing of the S-PDMS and the D-PDMS@BA prepolymers. Finally, the unstable D-PDMS@BA network was removed to obtain the HNP-PDMS@C by ultrasonic cleaning of the composite elastomer with ethanol. Two 1-mm thick S-PDMS sheets sandwiched a 1-mm thick HNP-PDMS@C to create the flexible sensor material.

### Characterizations

The pores were obtained using mercury injection (mercury intrusion porosimetry, Micromeritics AutoPore V 9620, USA). The surface morphology of the elastomers was characterized using a tungsten filament SEM (ZEISS EVO10, Germany). The chemical composition of the surface was obtained using an energy-dispersive spectrometer (EDS, OXFORD instruments, Abingdon, UK). The 3D topographic features of the polymer surfaces were examined using a laser scanning confocal microscope (Leica DCM8; Wetzlar, Germany). The roughness of every sample was estimated in the same scan sizes (objective lens 50×). An optical microscope (CX40M, Sunny Optical Technology Co. Ltd., Ningbo, China) equipped with a polaroid and warm white LED light (color temperature of 3000–3300 k) was used to observe the liquid–liquid phase separation. The truncated face channels and diffraction of the elastomers were observed using transmission electron microscopy (Talos L120CG2). FTIR was performed on a Thermo Scientific Nicolet iS50 in the attenuated total reflectance mode. SAXS was conducted using a small-angle X-ray scatterer (Anton Parr SAXS point 2.0) to characterize the structural changes in the samples with different elastomers. Uniaxial tensile and compressive strain tests were performed using a universal tensile-compression tester (ZQ 990, China). Dynamic mechanical analysis (DMA) was performed using a DMA equipment (Netzsch, Germany, DMA242) with a heating rate of 3 °C/min from −20 to 100 °C. An oscilloscope (Tektronix TBS2072B) was employed to capture and record the open circuit voltage. A LabVIEW-controlled digital source meter (Keithley 2400, America) recorded the resistance in real time.

### Oil absorption capability

For the oil injection, all HN-PDMS@BA samples were cut into 10 × 10 × 1 mm pieces and weighed to obtain the initial mass ($m_1$). Then, the mass of each HNP-PDMS@BA elastomer ($m_2$) was weighed to be obtained. Each HNP-PDMS@BA elastomer was injected with methyl silicone oil in a vacuum environment. After oil injection, the surface oil was removed using oil blotting paper, and the mass of oiled HNP-

PDMS@BA ($m_0$) was recorded (further details are presented in Section 2.2 of Supplementary Material). All experiments were repeated at least three times.

### Molecular dynamics simulations for heterogeneous network

MD simulations were performed using an initial monomer, followed by crosslinking, geometry optimization, energy minimization, and dynamics. All simulations were implemented using the Materials Studio software package via the Amorphous Cell and Forcite modules. The Dreiding force field[63] was used to determine atomistic interactions between polymer chains. Electrostatic interactions were computed using an Ewald representation, with temperature control achieved using a velocity-scale thermostat set to a relaxation time of 0.1 ps. Van der Waals interactions were calculated using an atom-based representation during model generation, equilibration, and subsequent production runs. The models were equilibrated at 300 K for 100 ps with a time step of 1 fs in the NVT ensemble, followed by a restart at 300 K for an additional 100 ps in the NVE ensemble to finalize the cohesive energy density calculations. The cohesive energy density results for S-PDMS and D-PDMS@BA were obtained by repeating these simulation steps. For multiple-chain models, MD simulations were equilibrated at either 300 or 398 K for at least 100 ps in the NVT ensemble. The MSD and radial distribution function of D-PDMS@BA and B atoms were subsequently analyzed based on these simulations.

### Molecular dynamics simulations for water wetting

All simulations were performed by LAMMPS. Water molecules were represented by the coarse-grained mW water molecules[64]. Water-solid interactions were described by 12-6 Lennard–Jones potential[65,66], and the wettability of solid surfaces was controlled by the solid-water interaction parameter ($\varepsilon$), while the cutoff distance for water–solid interaction was set to 8 Å. All wetting simulations were performed for 10 ns in NVT ensemble, at 300 K. The temperature was controlled with the Nose–Hoover thermostat with a relaxation time of 0.5 ps, and periodic boundary conditions were set. To explore the interaction between GCr15, water, and hydrophobic or hydrophilic surface, a simplified model was established (Supplementary Fig. 50) and each $\varepsilon$ was set (Supplementary Table 4). The Fe (001) plate model with a BCC lattice constant of 2.86 was built to simulate wetting the status of the steel (Supplementary Fig. 34), which shows approximately 78° of CA consistent with the previous study[54]. The hydrophobic or hydrophilic surface was built with an FCC lattice constant of 3.9, and the CA showed approximately 107° or 75° according to our previous study[67]. Images were visualized using the OVITO and Python packages in wetting simulation.

### Density functional theory calculations

All density functional theory calculations were performed using the Vienna Ab initio Simulation Package (VASP, version 6.3.0)[68]. The Exchange–correlation interactions were treated within the generalized gradient approximation, employing the Perdew–Burke–Ernzerhof functional[69]. Ion–electron interactions were modeled using the projector augmented-wave method[70]. A plane-wave cutoff of 520 eV was used for all calculations. For isolated molecules, a vacuum spacing of at least 15 Å was applied along all directions, and $\Gamma$-point sampling was used for Brillouin zone integration. Geometries were optimized until the residual atomic forces were below 0.02 eV Å$^{-1}$, with an electronic convergence criterion of 10$^{-5}$ eV. Electrostatic potentials were evaluated using VASP's default settings.

### Friction performance test

Friction tests were performed using a friction testing machine (UMT-2, USA) at a relative humidity of 40–50% and a temperature of approximately 25 °C. The upper sample was a GCr15 or Si3N4 ball with a

diameter of 9.5 mm. The lower sample was S-PDMS, P1, and P2 for the experiments. The loads of 2, 4, and 6 N were used; the reciprocating frequency was 4 Hz; The lubrication liquid was saltwater (3.5% of NaCl solution). The reciprocating stroke was 12 mm along the linear direction of the lower sample; hence, the average velocity was 48 mm/s. Before each friction test, the upper sample was ultrasonically cleaned with anhydrous ethanol for 5 min. Each combination was repeated at least thrice.

## Mechanical performance tests

Uniaxial tensile and compression tests were conducted on all samples at velocities of 5 and 1 mm/min, respectively. Cyclic tensile and compression tests were performed under the same conditions. Samples were initially stretched or compressed to a specific stretching or compression ratio and then immediately unloaded. Two unloading conditions were adopted: nominal stress unloading to 0 Pa and displacement unloading to 0 mm. To minimize the Mullins effect[71–73], the compression modulus was calculated within the linear region, ensuring a minimum stress of 0.015 MPa and linear fit with an $R^2 > 0.99$. Each experiment was repeated at least three times.

## Data availability

All data that support the findings of this study are included in the article and its Supplementary Information. The Supplementary Information data are available from the corresponding author upon request. Source data are provided with this paper.

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

## Acknowledgements

This work was supported by the Fundamental Research Fund of Xi'an Jiaotong University for Youth Innovation Team (xtr052024006 to L.Q.), the Natural Science Foundation of Hunan Province (2024JJ7028 to L.Q.), the National Natural Science Foundation of China (51975458 to L.Q.), and the Fundamental Research Funds of Xi'an Jiaotong University for Students (No. 11913224000055 to Z.W.). We thank Miss Hang Guo at instrument Analysis Center of Xi'an Jiaotong University for their assistance with CLSM and SEM analysis.

## Author contributions

Y.W. and L.Q. conceived the idea. Y.W., M.S., Z.W., S.L., Z.M., X.H., W.X., H.Y., and J.L. performed experiments. L.Q., X.G., S.Y., and G.D. supervised the project. Y.W. conducted the simulations. Y.W., L.Q., Z.M., S.L., and K.Y. analyzed and interpreted the data in collaboration with the other co-authors. Y.W. and L.Q. wrote the manuscript with the assistance of the other coauthors.

## Competing interests

The authors declare no competing interests.
