## [Transparent Peer review file · Nature Communications]

Borate Driven Heterogeneous Networks for Porous Elastomers with Improved Tribological and Mechanical Performances

Corresponding Author: Professor Liguo Qin

Version 1:

Reviewer comments:

Reviewer #1

(Remarks to the Author)

This article presents an innovative borate-driven approach for fabricating porous PDMS elastomers. By employing vacuum curing combined with controlled alcoholysis, the authors successfully achieve selective removal of one polymer network, resulting in interconnected porous structures with stabilized interfaces. The prepared materials demonstrate exceptionally low friction under saltwater lubrication conditions and enhanced mechanical strength, underscoring their significant potential for applications in water-lubricated components and flexible sensing devices. The authors thoroughly elucidate both the synthesis procedure and underlying mechanisms governing the material's formation. I believe this manuscript offers valuable contributions to the field and recommend its publication after the authors address the following questions.

1. The paper repeatedly emphasizes the stabilizing role of borate groups at the interface but primarily relies on molecular dynamics simulations and macro-level performance tests. It lacks direct characterization to confirm how borate bonds interact with the siloxane backbone, including bond strength and dynamic reversibility.
2. The conclusion section proposes potential applications in water-lubricated components and flexible sensors, which typically demand durability, fatigue life, and environmental stability. Have any relevant cyclic stretching or long-term wear tests ($>10^5$ cycles) been conducted to evaluate aging or performance degradation?
3. Although ultra-low friction and wear are achieved under saltwater lubrication, the long-term interaction and durability of corrosion products (iron-oxide nanoclusters) within the pore structure remain underexplored. For instance, does the hydrophilicity and nanocluster trapping diminish after repeated friction or prolonged saltwater immersion?
4. Following the previous point, how might this alter tribological behavior if corrosion products accumulate excessively or detach?
5. The authors claim this method suits various PDMS systems and potentially non-silicone elastomers. However, only limited evidence (e.g., Ecoflex@BA) is provided, which seems insufficient to substantiate this claim.

Reviewer #2

(Remarks to the Author)

Title: Borate-Driven Heterogeneous Networks for Porous Elastomers with Enhanced Tribological and Mechanical Performances

Author: Wu et al

The manuscript describes a method for making porous PDMS elastomer which have favorable tribological and mechanical properties. The method involves phase separation within the bulk of the crosslinkable mix and use of borate at the interface of the phases for stabilizing the two-phase system. Authors report also reduction in friction coefficient by over 95% in presence of salt water. Beside that, the porous material exhibits stretchability of 1250%. While the authors have presented in detail the method of making sample, they have not provided sufficient rational of several observations from the experiments. In other word, while the topic looks interesting, scientifically it does not appear very enriching yet. Furthermore, the manuscript is also not very easy to read. There are mix-up of tenses, long sentences, unnecessary adverbs, typographical errors. I have listed some of them below and have also written my comments and questions. Based on these observations I do not recommend acceptance of this manuscript for publication.

(a) There are typographical errors and mixing of tenses.

In page 3 line 71, "... presented ..." : why past tense?

In page 2 line 48, "Friction will destroy ..." why future tense?

In page 4 line 81, "... rater ..." : typo.

In page 4 line 81-83 "Unique advantages ... borate sites", not clear what the authors want to say here. What is "universal method"?

(b) line 83-86: Author claims making "highly specialized ...". An interesting material has been made but it is not clear what is highly special about it. For example, pores do not appear interconnected here. Are they? Authors don't even mention the pore diameter. Author has mentioned surface roughness attained at different temperature and pressure. But how was the roughness estimated? RMS roughness is expected to be different at different scan sizes. Furthermore, since the pores are not connected, is it fair to say that it is only the surface pores which could directly influence the friction coefficient; the pores that supposedly formed at the bulk of the material didn't have any effect.

(c) In fact, in figure 3C to E the symbols used for the data points are so small that it can not be seen if there is any error bar there. Furthermore, it is not clear why the slopes of these lines are different and what exactly that signifies. It is not clear also what the intercepts in these plots mean.

(d) Page 5 line 108-109: "In the ...S-PDMS", Not clear what the author wants to say here.

(e) Page 5 line 114: "...proposed ..." Should it be past tense here?

(f) Page 9 line 181-183 "As the stretch ...pronounced" This sentence needs rewriting.

(g) Page 12 line 264: "The particle ...being isolated" This statement is not correct. It is possible to make network of interconnected pores using particle templates. Secondly, the schematic of figure 4A and the SEM images of figure 4B that the authors have presented show that the pores are like vertical holes that spans from one cross-section to other but they do not appear to be interconnected.

(h) Page 13 line 281: "Under certain ...COF": What are these "certain wetting conditions"? Why do circular textures (I understand presence of holes) at all reduce the COF?

(i) Page 13 line 281: "Wetted surfaces ... in methods)" What is the meaning of this statement? ... particularly that of the word "guarantee"?

(j) Page 14 line 293 "... water film density ..." this phrase may have some meaning in MD solution, but practically it either doesn't have meaning or it signifies some different parameter.

(k) Page 14 line 297 "Amazingly, COF ..." what is so "amazing" about it. Such adverbs are best avoided in a scientific document! Authors have rationalized this observation by attributing it to generation of debris because of "wear". Authors have presented also debris volume for different cases. Shouldn't the debris volume continue to increase with time during of the sliding experiment? Shouldn't the debris volume vary with the speed of sliding? In fact, authors do not mention the sliding speed at all in the manuscript.

(l) In figure 5D, authors have presented the COF as a function of reciprocal cycles. What is the significance of carrying out this experiment over 12000 cycles. Why the COF values for all cases become different after 4000 cycles. Secondly even in one cycle the normal load and shear load and consequently the COF should exhibit fluctuations. Authors do not make any mention of that. Secondly did the authors obtain shear load at different normal load?

(m) Page 14 line 309: "... (Figure 22C)...", There is no such figure!

(n) The stretch percentage for some of the samples were found to be quite high, as high as 1250%. What leads to this increase, low crosslinking density, viscoelastic effect, smaller physical entanglement, what? Authors have not thrown any light on the possible mechanism that leads to this increase in extensibility.

Reviewer #3

(Remarks to the Author)

The authors report a novel, template-free emulsion method to create purely porous silicone elastomers by driving heterogeneous network formation using borate-based dynamic crosslinkers. By combining a borate-modified PDMS precursor with a commercial PDMS prepolymer and inducing spontaneous phase separation under controlled thermal and vacuum conditions, they generate an interconnected porous structure after selectively removing the sacrificial borate phase. The resulting materials exhibit remarkably low friction and wear under aqueous lubrication while retaining excellent stretchability and toughness. Furthermore, the approach allows precise tuning of pore morphology and mechanical properties through processing parameters and can be extended to other silicone elastomer systems, demonstrating broad potential for applications in low-friction, wear-resistant soft materials.

However, the manuscript, in its current state, may not meet the standards for publication in Nature Communications, a journal known for its groundbreaking and influential contributions to the scientific community. The idea and concept of the

paper are excellent and novel, but the available data on this particular aspect appears to be insufficient. Furthermore, there is a need for more comprehensive supporting evidence and data to bolster the proposed hypotheses.

The authors are encouraged to consider the following suggestions for enhancing the manuscript in preparation for future submissions.

To strengthen the manuscript, the authors should consider (Major):

1. To fulfill the objectives proposed by the authors, additional stability and wear-rate measurements under thermal, high-temperature, UV, and aging conditions must be conducted.
2. Beyond the current stability tests over several thousand cycles, experiments addressing a much longer fatigue life are required. Considering the applications the authors propose, stability over only a few thousand cycles is insufficient.
3. The authors have validated their method in PDMS and Ecoflex. The claim that the fabrication approach applies to all polysiloxane systems is unsupported by evidence from only two examples; additional validation cases are needed to substantiate this assertion.
4. The authors' scientific novelty and extensive material analyses, including modeling, are highly commendable; demonstrating a practical application is necessary to better align with the scope of Nature Communications. Any example—such as the machinery, robots, bearings, or biological models—would be suitable.

Version 2:

Reviewer comments:

Reviewer #1

(Remarks to the Author)

The authors have fully addressed my concerns. I suggest the publication of the revised manuscript.

Reviewer #2

(Remarks to the Author)

[In correspondence with the editors, the referee expressed that the manuscript is ready for publication as-is.]

Reviewer #3

(Remarks to the Author)

The authors present their research in a logical and coherent manner, supported by robust experimental design, sound methodology, and convincing data analysis. The novelty of the work is evident in the way it bridges current knowledge gaps and provides new insights that are likely to stimulate further research and practical applications. I find no substantive weaknesses that would require revision, and I am pleased to recommend the manuscript for acceptance in its current form.

Response Letter

Thank you for the reviewers' insightful comments and suggestions, which have been extremely helpful in improving our work. We have studied comments carefully and tried our best to made corrections to meet the approval. In addition, we have modified some formats and sentences to express the meaning clearly and accurately. Point-by-point responses to each of the Reviewer comments are listed in the following content. [Referees' comments are in black; Author responses are in blue; The original texts are in purple; Revisions in the manuscript are highlighted.]

Reviewer #1 (Remarks to the Author):

This article presents an innovative borate-driven approach for fabricating porous PDMS elastomers. By employing vacuum curing combined with controlled alcoholysis, the authors successfully achieve selective removal of one polymer network, resulting in interconnected porous structures with stabilized interfaces. The prepared materials demonstrate exceptionally low friction under saltwater lubrication conditions and enhanced mechanical strength, underscoring their significant potential for applications in water-lubricated components and flexible sensing devices. The authors thoroughly elucidate both the synthesis procedure and underlying mechanisms governing the material's formation. I believe this manuscript offers valuable contributions to the field and recommend its publication after the authors address the following questions.

Response:

We highly appreciate Reviewer #1's valuable and helpful comments. We have done our best to provide appropriate responses and the revised manuscript to address these questions.

1. The paper repeatedly emphasizes the stabilizing role of borate groups at the interface but primarily relies on molecular dynamics simulations and macro-level performance tests. It lacks direct characterization to confirm how borate bonds interact with the siloxane backbone, including bond strength and dynamic reversibility.

Response:

Thank you for the valuable comment. To address your concern, we added Fourier transform infrared (FTIR) spectroscopy characterization, its analysis to confirm the bond strength and dynamic reversibility. As far as we know, direct characterization of molecular skeleton interactions is very difficult, but we have used density functional theory (DFT) calculations with higher accuracy than molecular dynamics simulations.

We hope these meet your requirements. According to the previous study (Chem. Mater. 2014. 26. 12, 3781-3795), **boroxine** will be generated after the reaction between PDMS and boric acid, which is verified by the result of FTIR as shown in Supplementary Fig. 2A. Boroxanes with multiple boronate esters support the speculation of B atom enriched at the two-phase boundary. Based on the wavelengths (ν) of Si-O and B-O displayed by FTIR, the force constant (k) can be calculated according to the following formula:

$$\nu = \frac{1}{2\pi c} \sqrt{\frac{k}{\mu}}$$

c and μ are speed of light and reduced mass. Substituting the wavelength values of Si-O and B-O measured by FTIR into the above equation, we can get k_{B-O} (677 N/m) > k_{Si-O} (606 N/m). Using a repeating unit in D-PDMS@BA as a model, results of DFT calculation show that the bond dissociation energies of B-O and Si-O are 6.05 eV and 5.65 eV ($BDE_{B-O} > BDE_{Si-O}$), respectively. The bond dissociation energy results are consistent with the force stiffness results. Therefore, **bond strength** of B-O is more than that of Si-O. Based on the above molecular model, we calculated the electrostatic potential of the molecule as shown in the Supplementary Fig. 3 to intuitively display **interactions of bonds with the siloxane backbone**. When boric acid and HPDMS are cross-linked, a distinct repulsive region is formed. From the electrical potential values, compared with the other groups, the silanol group is more likely to be closer to the boric acid; The non-polar methyl groups and polar hydroxyl groups are less likely to attract each other. To investigate the **dynamic reversibility**, we performed FTIR tests on D-PDMS@BA ethanol solution and D-PDMS@BA after evaporation of ethanol. FTIR result in Supplementary Fig. 2B displays that the B-O and C-O peaks of D-PDMS dissolved in ethanol were broadened and shifted, respectively. The FTIR result of D-PDMS after evaporation of ethanol are consistent with that of D-PDMS not dissolved in ethanol. These results illustrate the dynamic reversibility of D-PDMS under the influence of ethanol. Besides, in our earlier study (*ACS Applied Materials & Interfaces*, 2024, 16(43): 58980-58990), we demonstrated its force-triggered dynamic reversibility.

Revised or added Figure:

Supplementary Fig. 2. (A) Fourier transform infrared spectrum of S-PDMS, D-

PDMS@BA, HN-PDMS@BA, and HNP-PDMS@BA. (B) Fourier transform infrared spectrum of ethanol, the mixture of D-PDMS@BA and ethanol, and D-PDMS@BA after ethanol evaporated.

Supplementary Fig. 3. Electrostatic potential map of a repeating unit in D-PDMS@BA

Revised or added text (main text, page 5):

In the phase-separation mixture, as shown in Supplementary Fig. 2, all hydroxyl groups of H-PDMS reacted after curing to obtain HN-PDMS@BA. Fourier transform infrared spectroscopy (FTIR) results show that the borate group of D-PDMS@BA can be considered the distinguishing characteristic, compared to S-PDMS (for detailed discussion, see Supplementary Information, Section 2.1). Consequently, we considered the interaction between borate ester and silicone oil as an important factor of incompatibility. The electrostatic potential of the repeating unit in D-PDMS@BA was calculated as shown in the Supplementary Fig. 3, to intuitively display the interactions of the borate bond with the siloxane backbone. The interactions suggest that the incompatibility is likely due to the repulsion between the Si-O bonds and the oxygen in the Si-O backbone (for detailed discussion, see Supplementary Information, Section 2.1).

Revised or added text (Supplemental Information, Section 2.1):

Based on the wavenumbers (ν) of B-O and Si-O displayed by FTIR, the force constant (k) can be calculated according to the following formula:

$$\nu = \frac{1}{2\pi c} \sqrt{\frac{k}{\mu}} \quad (S1)$$

where c and μ are speed of light and reduced mass. Because of k_{B-O} (677 N/m) > k_{Si-O} (606 N/m), bond strength of B-O is more than that of Si-O. Then, we evaluated bond dissociation energies (BDEs) for two chemically distinct bonds within the target molecule, Si-O (siloxane bridge) and B-O (the terminal B-O-Si linkage). To avoid open-shell radicals and ensure closed-shell fragments (Frag), we employed a hydrogenation (H₂-capping) scheme that conserves atom counts:

For a generic A–B bond in the parent molecule M, define the reaction

where each fragment is constructed by cleaving A–B and adding one H atom to A and one H atom to B to satisfy valence. The bond dissociation energy is then

$$EBDE(A - B) = E[\text{Frag}(A - H)] + E[\text{Frag}(B - H)] - E[M] - E[H_2] \quad (\text{S2})$$

(i) Si–O (siloxane) bond inside the chain)

Cut the selected Si–O bridge in HO- [Si (CH₃)₂-O]₆-B(OH)₂, yielding two fragments: the Si-terminated part is capped to –Si (CH₃)₂-H, and the O-terminated part is capped to –O–H (silanol). Apply Eq. (1):

$$EBDE(\text{Si} - \text{O}) = E[\text{Frag with Si} - \text{H}] + E[\text{Frag with O} - \text{H}] - E[M] - E[H_2] \quad (\text{S3})$$

(ii) B–O (terminal B–O–Si linkage)

Cleaving the B–O bond that connects B(OH)₂ to the chain gives (a) the chain end capped to –Si–O–H and (b) a boron fragment HB(OH)₂ (i.e., B gains one H). The corresponding BDE is

$$EBDE(B - O) = E[\text{Si} - \text{O} - \text{H terminated chain}] + E[\text{HB}(\text{OH})_2] - E[M] - E[H_2] \quad (\text{S4})$$

This H₂-capping protocol yields closed-shell fragments and is robust for siloxane and boron–oxygen bonds. For nonequivalent Si–O bonds along the chain (e.g., near termini vs. interior), Eqs. S2 and 3 were applied to each distinct local environment. Based on the above method, the calculation results by density functional theory show that the BDEs of B–O and Si–O are 6.05 eV and 5.65 eV, respectively (For more details on the modifications to this section, see the Support Information).

Revised or added text (main text, Method, page 25):

Density functional theory calculations

All density functional theory (DFT) calculations were performed using the Vienna Ab initio Simulation Package (VASP)⁶¹. Exchange–correlation interactions were described within the generalized gradient approximation using the Perdew–Burke–Ernzerhof functional⁶², and ion–electron interactions were treated with the projector augmented-wave method⁶³. A plane-wave cutoff of 520 eV was used. For isolated molecules, a vacuum spacing of at least 15 Å was applied along all directions and Γ -point sampling was adopted. Geometries were relaxed until the residual atomic forces were below 0.02 eV Å⁻¹, with an electronic convergence threshold of 10⁻⁵ eV. Unless otherwise stated, electrostatic potentials were evaluated using VASP’s default settings.

2. The conclusion section proposes potential applications in water-lubricated components and flexible sensors, which typically demand durability, fatigue life, and environmental stability. Have any relevant cyclic stretching or long-term wear tests (>10⁵ cycles) been conducted to evaluate aging or performance degradation?

Response:

Thank you for the significant comment. We performed the long-term wear test (>1.2×10⁵ cycles) for P1-G. With reciprocating cycles progressing and water evaporating, the COF increases, but the COF quickly decreases to a lower level after adding a little salt water (~0.1 mL) as shown in Supplementary Fig. 33. Besides, P1 was stretched for over 2000 cycles, with a stretch ratio of 20% in each cycle. Then,

friction testing of the sample after cyclic stretching revealed that it still exhibited low friction, with the average COF less than 0.15 (Supplementary Fig. 35). Furthermore, after 3 weeks of salt water immersion, UV irradiation, and heating treatment, P1 can stably reduce the COF. An increase in the COF at 60 °C after 10000 cycles, which is due to the evaporation of water. However, the low COF state can be recovered after adding salt water (Supplementary Fig. 33). These indicate that this porous surface has the potential for long-term service.

Revised or added Figure:

Supplementary Fig. 33. The COF–cycle curve of P1-G after 122400 reciprocating cycles (Yellow area: Lubrication deteriorated as the water evaporated; Blue area: Lubrication improved after adding 0.1 mL of salt water).

Supplementary Fig. 35. (A) The COF–cycle curve and (B) average COF of P1-G after aging or degradation treatment.

3. Although ultra-low friction and wear are achieved under saltwater lubrication, the long-term interaction and durability of corrosion products (iron-oxide nanoclusters) within the pore structure remain underexplored. For instance, does the hydrophilicity and nanocluster trapping diminish after repeated friction or prolonged saltwater immersion?

Response:

Thank you for the constructive suggestions. We repeated the friction test of P1-G three times in situ without processing and found that COF reduction still occurred (Supplementary Fig. 37). After repeated friction, the CA is still close to that after the first friction (Supplementary Figs. 23C and D). The corresponding SEM and EDS results of the worn surface are shown in Supplementary Fig. 34B, which reveals the presence of wear debris and iron in the pores, indicating that the nanoclusters were still trapped. Besides, combining the COF and SEM results of the surface subjected to 122400 cycles in (Supplementary Fig. 34A), it can be seen that the function of surface pores in capturing wear debris is still effective. Furthermore, the long-term NaCl immersion (3 weeks) further reduces the COF (Supplementary Fig. 35).

Revised or added Figure:

Supplementary Fig. 37. After a 12000-cycle friction test, the friction test was repeated without any treatment.

Supplementary Fig.23. The wetting status of water droplets on the (A) S-PDMS surface, (B) P1 surface, (C) P1 surface after one friction experiment, and (D) P1 surface

after three repeated friction experiments.

Supplementary Fig. 34. Results of SEM and EDS elemental analysis after (A) 122400 reciprocating cycles and (B) three consecutive friction tests.

Supplementary Fig. 35. (A) The COF–cycle curve and (B) average COF of P1-G after aging or degradation treatment.

Revised or added text (Supplemental Information, Section 2.3):

To investigate the stability of the low-friction transition and low-wear state, we perform fatigue and aging tests on the P1 (HNP-PDMS@BA-125/-0.1MPa). As shown in Supplementary Fig. 33, with reciprocating cycles progressing and water evaporating, the COF increases, but the COF quickly decreases to a lower level after adding a little (~0.1 mL) salt water. SEM and EDS results show that the pores of P1 are always able to capture the wear debris during this process (Supplementary Fig. 34A). COF results of other tests are shown in Supplementary Fig. 35. Friction testing of the sample after cyclic stretching (over 2000 cycles, with a stretch ratio of 20% in each cycle) revealed that it still exhibited low friction, with the average COF less than 0.15. Furthermore, after 3 weeks of salt water immersion, UV irradiation (365 nm), and heating treatment (60 °C), P1 can stably reduce the COF. An increase in the COF at 60 °C after 10000 cycles, which is due to the evaporation of water. However, the low COF state can be recovered after adding salt water.

4. Following the previous point, how might this alter tribological behavior if corrosion

products accumulate excessively or detach?

Response:

We are very grateful for the comments. Following the Response to 2, as shown in Supplementary Fig.33, the yellow area displays the change in COF when corrosion products accumulate by causing the water to evaporate almost completely. This shows that excessive accumulation leads to an increase in COF, but the COF can drop rapidly when the products are diluted. The wear rate of the process is less than P1-G (Supplementary Fig.36). To study the effect of product detachment on friction, after the first friction test, the mixed liquid on the surface was wiped off by the dust-free fabric, and the new friction test was carried out. The COF dropped rapidly with almost no running-in period of thousands of cycles (Supplementary Fig.38). Compared with S-PDMS-G, P1-Si, and P2-Si, the COF of P1-Si still dropped by over 60%. We speculate that the old ball with products still works. To verify that, we changed the new ball after the mixed liquid on the surface was wiped off by the dust-free fabric and performed a friction test. The friction results show that a running-in period of thousands of cycles is necessary under this condition (Supplementary Fig.39).

Revised or added Figure:

Supplementary Fig. 33. The COF–cycle curve of P1-G after 122400 reciprocating cycles (Yellow area: Lubrication deteriorated as the water evaporated; Blue area: Lubrication improved after adding 0.1 mL of salt water).

Supplementary Fig. 36. P1-G-8.5h represents the sample after 122400 cycles of friction.

Supplementary Fig. 38. After the first friction test, wipe off the surface mixture without changing the ball: repeat the test.

Supplementary Fig. 39. After the first friction test, wipe off the surface mixture and replace the ball with a new one: repeat the test.

Revised or added text (Supplemental Information, Section 2.3):

In any case, the wear rate of the aging sample ($< 6.69 \times 10^{-5} \text{ mm}^3/\text{N}\cdot\text{m}$) was reduced by over 75% compared with the wear rate of S-PDMS-G, as shown in Supplementary Fig. 36. In particular, when the friction time was extended to 8.5 h (122400 cycles), the wear rate was further reduced to as low as $\sim 8.62 \times 10^{-6} \text{ mm}^3/\text{N}\cdot\text{m}$. We repeated the friction test of P1-G three times in situ without processing and found that COF reduction still occurred (Supplementary Fig. 37). After repeated friction tests, the CA is still close to that after the first friction test (Supplementary Figs. 23C and D). The corresponding SEM and EDS results of the worn surface are shown in Supplementary Fig. 34B, which reveals the presence of wear debris and iron in the pores, indicating that the nanoclusters were still trapped. To study the effect of product detachment on friction, after the first friction test, the mixed liquid on the surface was wiped off by the dust-free fabric and the new friction test was carried out. The COF dropped rapidly with almost no running-in period of thousands of cycles (Supplementary Fig. 38). Compared to S-PDMS-G, P1-Si, and P2-Si, the COF of P1-Si still dropped by over 60%. We infer that the old ball with products still works. To verify that, we changed the new ball after the mixed liquid on the surface was wiped off by the dust-free fabric and performed a repeated friction test. The friction results show that a running-in period of thousands of cycles is necessary under this condition (Supplementary Fig. 39). Overall, all these indicate that this porous surface has the potential for long-term service.

5. The authors claim this method suits various PDMS systems and potentially non-silicone elastomers. However, only limited evidence (e.g., Ecoflex@BA) is provided, which seems insufficient to substantiate this claim.

Response:

Thank you for pointing out the potential issue, and this is an important comment. To address the issue, we added three other elastomer materials (DC170, DC527, and RTV615) for validation. Using the same preparation method as HNP-PDMS, we successfully prepared new porous elastomers (DC170@BA, DC527@BA, and RTV615@BA) as shown in Supplementary Fig. 46. These porous elastomers have the potential to improve the elongation.

Revised or added Figure:

Supplementary Fig. 46. The surface SEM images of (A) DC170, (B) DC170@BA, (D) DC527, (E) DC527@BA, (G) RTV615, and (H) RTV 615@BA. Stress-stretch curves of (C) DC170 and DC170@BA, (F) DC527 and DC527@BA, (J)RTV615 and RTV615@BA.

Revised or added text (main text, Section 2.6):

The introduction of D-PDMS@BA significantly delayed the fracture of the elastomer, and the toughness increased to 150 %, despite the elastomer being porous. Furthermore, we successfully prepared new porous elastomers using three other PDMS materials (DC170, DC527, and RTV615), as shown in Supplementary Fig. 46. These porous elastomers also exhibited elongation improvement potential. These results proved that our method was applicable to the preparation of mechanically enhanced porous elastomers for all PDMS systems.

Reviewer #2 (Remarks to the Author):

Title: Borate-Driven Heterogeneous Networks for Porous Elastomers with Enhanced Tribological and Mechanical Performances

Author: Wu et al

The manuscript describes a method for making porous PDMS elastomer which have favorable tribological and mechanical properties. The method involves phase separation within the bulk of the crosslinkable mix and use of borate at the interface of the phases for stabilizing the two-phase system. Authors report also reduction in friction coefficient by over 95% in presence of salt water. Beside that, the porous material exhibits stretchability of 1250%. While the authors have presented in detail the method of making sample, they have not provided sufficient rational of several observations from the experiments. In other word, while the topic looks interesting, scientifically it does not appear very enriching yet. Furthermore, the manuscript is also not very easy to read. There are mix-up of tenses, long sentences, unnecessary adverbs, typographical errors. I have listed some of them below and have also written my comments and questions. Based on these observations I do not recommend acceptance of this manuscript for publication.

Response:

We are very grateful to Reviewer #2 for the comments. Based on your comments and questions, we have improved our manuscript. We hope that the readability and scientific quality of the revised manuscript have been sufficiently improved and meet your requirements.

(a) There are typographical errors and mixing of tenses.

In page 3 line 71, "... presented ..." : why past tense?

In page 2 line 48, "Friction will destroy ..." why future tense?

In page 4 line 81, "... rater ..." : typo.

In page 4 line 81-83 "Unique advantages ... borate sites", not clear what the authors want to say here. What is "universal method"?

Response:

We appreciate the reviewers' constructive comments and are deeply sorry for confusing you. We reviewed the manuscript again and corrected the tense errors and typos. For the final question, "Universal method" means that our method can be applied to a variety of PDMS systems, and a variety of boric acid derivatives can work. The abrupt appearance here might be confusing, so we've changed the way this sentence is written. For details on the changes, please see the revised text section below.

Revised or added text (main text, page 3, line 71):

This study presents a method for preparing heterogeneous-network porous PDMS (HNP-PDMS) elastomers, as shown in Fig. 1A (For all abbreviations and their expansions, see Supplementary Table S1).

Revised or added text (main text, page 2 line 48):

Friction can destroy the passivation film on the metal surface and promote corrosion.

Revised or added text (main text, page 4, lines 81-83):

Mechanical performance advantages could be imparted to the porous elastomers, such as a stretch percentage of 1250%, by controlling the temperature, pressure, and borate sites (Fig. 1C). Using a wide variety of BA derivatives and silicone materials, this study offers a new approach for designing specialized porous elastomers for enhanced water lubrication performance and preparing flexible sensors.

(b) line 83-86: Author claims making “highly specialized ...”. An interesting material has been made but it is not clear what is highly special about it. For example, pores do not appear interconnected here. Are they? Authors don't even mention the pore diameter. Author has mentioned surface roughness attained at different temperature and pressure. But how was the roughness estimated? RMS roughness is expected to be different at different scan sizes. Furthermore, since the pores are not connected, is it fair to say that it is only the surface pores which could directly influence the friction coefficient; the pores that supposedly formed at the bulk of the material didn't have any effect.

Response:

We are very grateful for your valuable suggestions. 1) In fact, the pores are interconnected. As shown in Supplementary Fig. 19, the cross-section also has pores. After oil is injected, the oil can flow out of the cross-section. Additionally, the results of transmission electron microscopy (Supplementary Fig. 20) and laser confocal microscopy (Fig. 4D) suggest the presence of internally connected channels. These further confirmed the interconnected structure. In the previous manuscript, we discussed the surface pores on page 11, lines 221-226, and internal pores on page 12, lines 248-258. To avoid unclear expressions, we have modified the sentences (see the following revised text). 2) The roughness (R_q /RMS) of each sample was estimated using the same scan sizes (objective lens 50 \times) with laser confocal microscopy. The S_q in Fig. 4F is the average of 3 different samples cured at the same conditions. S_q also describes the surface porosity from another perspective. 3) In this study, the pores are interconnected, but whether or not they are interconnected does not directly affect the COF. The presence of internal pores may affect the modulus and, therefore, the COF. As shown in Fig. 5D and Fig. 6B, although the moduli of P1(HNP-PDMS@BA/-0.1) and P2 (HNP-PDMS@BA/-0.0) vary significantly, the difference in their COF is not significant. However, the presence of GCr15 induces in-situ hydrophilic modification of the P1 and P2 surface under corrosion, reducing the COF. Therefore, the surface pores contribute to the low COF (both by capturing wear debris, improving lubrication, and by accelerating corrosion under high roughness) rather than internal pores.

Revised or added text (main text, page 11, lines 225-226)

Conversely, when the pressure was improved (-0.1 to 0 MPa), the pores became smaller (5.16 to 1.16 μm) and sparser (Fig. 4C and Supplementary Fig. 14).

Revised or added text (main text, page 12, lines 248-258)

Subsequently, to investigate the internal microscopic pore structures of the porous elastomers, the internal morphology of the phases in HNP-PDMS@BA was observed using confocal laser scanning microscopy because of its transparency (Fig. 4D), revealing an internal interconnected and interpenetrating structure. For further investigations of the internal framework, the porous elastomer was cut diagonally and observed using SEM (Supplementary Fig. 16B). When the HNP-PDMS@BA elastomer, filled with silicone oil, was cut, the oil flowed out of the pores within 30 s (Supplementary Fig. 19). To investigate the smaller-scale features, the nanochannels in the elastomer were analyzed using transmission electron microscopy. These nanochannels, resembling cylindrical pipes with diameters less than 500 nm, extended through the elastomer (Supplementary Fig. 20). Additional pore details are shown in Supplementary Fig. 21. At an air pressure of -0.1 MPa, HNP-PDMS@BA primarily formed micron-sized pores (>0.5 μm) and mesopores (<50 nm), while at a pressure of 0.0 MPa, it formed more macropores (>50 nm). The absorption of silicone oil (30–50%) was significantly higher than that of mercury (porosity of approximately 10%), which may be attributed to the swelling of HNP-PDMS@BA owing to oil absorption^{23,51}. These findings suggested that HNP-PDMS@BA featured a microsphere airspace and interconnected micro–nano channels.

Revised or added text (main text, page 22, lines 466-468)

The 3D topographic features of the polymer surfaces were examined using a laser scanning confocal microscope (Leica DCM8; Wetzlar, Germany). The roughness of every sample was estimated in the same scan sizes (objective lens 50 \times).

Revised or added text (Supplementary Discussion, Section 2.2)

The Sq in the Fig. 4F is the average of 3 different samples cured under the same conditions.

(c) In fact, in figure 3C to E the symbols used for the data points are so small that it can not be see if there is any error bar there. Furthermore, it is not clear why the slopes of these lines are different and what exactly that signify. It is not clear also what the intercepts in these plots mean.

Response:

We are very sorry that you feel confused. In fact, these points are the average radii of gyration (R_g) calculated by Supplementary Eq. S10. We have added error bars. In this study, we mainly utilized two types of PDMS networks. The introduction and partial exfoliation of D-PDMS caused changes in the molecular network, which in turn led to different slopes. By calculating the tensile modulus, we found that the slope and modulus are surprisingly consistent in value. Therefore, the slope can represent the strength and stiffness of the molecular chain. The intercepts are the R_g of the elastomer at 0% stretching after fitting.

Revised or added Figure:

Fig. 3. Gyration radius (R_g) as a function of the stretching ratio for (C) S-PDMS, (D) HN-PDMS@BA, and (E) HNP-PDMS@BA. The intercepts are the R_g of the elastomer at 0% stretching after fitting.

Revised or added text (main text, page 9, lines 190-193):

Notably, the slope values of the fitted lines for S- and HNP-PDMS are numerically similar to the tensile modulus (as discussed in Section 2.4), demonstrating that the slope values can represent the strength and stiffness of the molecular chain.

(d) Page 5 line 108-109: “In the ...S-PDMS”, Not clear what the author wants to say here.

(e) Page 5 line 114: “...proposed ...” Should it be past tense here?

(f) Page 9 line 181-183 “As the stretch ...pronounced” This sentence needs rewriting.

Together the response:

We appreciate the reviewer's careful reading and have revised the sentences accordingly.

Revised or added text (main text, page 5, lines 108-109):

In the phase-separation mixture, as shown in Supplementary Fig. 2, all hydroxyl groups of H-PDMS reacted after curing to obtain HN-PDMS@BA. Fourier transform infrared spectroscopy (FTIR) results show that the borate group of D-PDMS@BA can be considered the distinguishing characteristic, compared to S-PDMS.

Revised or added text (main text, page 5, line 114):

Consequently, we consider that increasing the D-PDMS@BA content (or the borate group content) intensifies the repulsion between D-PDMS@BA and S-PDMS, thereby enhancing the phase separation.

Revised or added text (main text, page 9, lines 181-183):

We stretched the elastomers from 0 to 80% and observed the SAXS patterns during this process (Fig. 3A). As the stretch increased, the isotropy was disturbed, and the molecular chains became oriented toward the direction of stretching. Among the three kinds of samples (S-PDMS, HN-PDMS@BA, and HNP-PDMS@BA), the anisotropy of HNP-PDMS@BA was most pronounced.

(g) Page 12 line 264: “The particle ...being isolated” This statement is not correct. It is

possible to make network of inter-connected pores using particle templates. Secondly, the schematic of figure 4A and the SEM images of figure 4B that the authors have presented show that the pores are like vertical holes that spans from one cross-section to other but they do not appear to be interconnected.

Response:

Thank you for the comments. 1) We agree with the reviewer's point. Adjusting the particle concentration and dispersion allows for the network of interconnected pores. However, this can further degrade the material's mechanical properties. Our statement was not accurate enough, so we have revised the sentence from the perspective of mechanical properties. The pores prepared by the particle template method are larger than those prepared by the emulsion method, which may lead to limited elongation at break and toughness. 2) To avoid misunderstanding, we modified Fig. 4A as follows. For detailed responses to interconnected pores, please refer to question (b).

Revised or added Figure:

Fig. 4. (A) Schematic diagram illustrating the control of surface pore morphology. The inset optical images show HNP-PDMS@BA before and after oil absorption.

Revised text (main text, page 12, lines 264-265):

The pores in the samples prepared by the particle template method were larger than those in the samples prepared by our emulsion method; larger pores were likely to lead to limited elongation at break and reduced toughness.

(h) Page 13 line 281: “Under certain ...COF”: What are these “certain wetting conditions”? Why do circular textures (I understand presence of holes) at all reduce the COF?

Response:

Thank you for the comments. 1) Generally, improving the hydrophilicity of the surface can reduce the COF, which is attributed to reduced slip of the lubricating liquid, which in turn improves lubrication conditions. This hydrophilization can cause the lubrication state to shift from boundary lubrication to mixed lubrication. 2) Surface texturing is a common method for reducing friction and wear. Over the past decade, our group has

conducted extensive experiments and theoretical calculations to understand why surface textures reduce COF. (Such as, *Friction*, 2025, 13(3). *Tribology International*, 2025, 212: 110976. *Tribology International*, 2023, 180: 108310. *Tribology International*, 2020, 149: 105733. *Applied Surface Science*, 2013, 268: 79-86.) In fact, circular textures do not necessarily reduce the COF. However, the circular texture is isotropic and easier to manufacture than other textures. It is generally believed that the reason for reducing the COF is that the structure can maintain a good lubrication state of the system, and hydrodynamic lubrication is formed near the pores. Once this lubrication is formed, it will be an effective means of reducing the COF. We apologize for the lack of clarity in this sentence and have therefore rewritten it. Besides, to facilitate the understanding of readers in multiple fields, we replace “texture” with “structure”.

Revised text (main text, page 13, line 281):

In particular, if the surface wettability is improved and combined with an appropriate surface structure, a transition of the lubrication regime can lead to a decrease in the COF^{52,53}.

(i) Page 13 line 286: “Wetted surfaces ... in methods)” What is the meaning of this statement? ... particularly that of the word “guarantee”?

Response:

We appreciate the reviewer's comment and have rewritten the sentences accordingly.

Revised text (main text, page 13, line 286):

In the MD simulations, the CAs of the surfaces were set such that the error in each case was within 3°, compared to the experimental CAs (see the Methods section for the details).

(j) Page 14 line 293 “... water film density ...” this phrase may have some meaning in MD solution, but practically it either doesn't have meaning or it signifies some different parameter.

Response:

We are deeply sorry that this sentence may have confused you. We have rewritten the sentence.

Revised text (main text, page 14, line 293):

Compared to the hydrophobic surface, a complete water film was formed between the ball and the hydrophilic surface. Correspondingly, the water density distribution is shown in Fig. 5C. It shows that when only the wetting effect is considered, hydrophilic surfaces are more likely to cause water to accumulate at the bottom of the ball than hydrophobic surfaces. This hydrophilic shift is likely to lead to a COF close to that of previous studies^{23,52}.

(k) Page 14 line 297 “Amazingly, COF ...” what is so “amazing” about it. Such adverbs are best avoided in a scientific document! Authors have rationalized this observation by attributing it to generation of debris because of “wear”. Authors have presented also debris volume for different cases. Shouldn’t the debris volume continue to increase with time during of the sliding experiment? Shouldn’t the debris volume vary with the speed of sliding? In fact, authors do not mention the sliding speed at all in the manuscript.

Response:

We appreciate the comments. 1) We have revised the adverb that needs to be avoided. 2) In the previous manuscript, we concluded that the decrease in COF was due to the transformation of the boundary lubrication into mixed lubrication (Figure 5I). The transformation is the reason for the decrease in the COF according to the *Stribeck Curve*. 3) The one purpose of our study is to investigate the effects of lubrication state changes induced by different friction pairs on wear rate and volume at low speeds. We don't seem to present debris volume, but rather wear volume and rate. The reviewer may be concerned about the display of wear. Wear rate and volume are important indicators to characterize the wear resistance of materials (*ASTM G133-22*). Wear volume generally continues to increase with time, but wear volume is calculated based on the product of sliding distance and wear scar cross-sectional area. The sliding distance follows as:

$$X = 0.002 \times t \times f \times L$$

$$N = t \times f$$

where:

X = total sliding distance of the ball, m,

N = number of cycles in the test,

t = test time, s,

f = oscillating frequency, Hz (cycles/s), and

L = length of stroke, mm.

According to the *Archard equation*, at low speeds, speed changes have little effect on hardness and thus wear rate. Therefore, we believe that the current sliding speed can demonstrate the potential application of our material, and we have also added the new result of over 120000 cycles to support the potentiality (See Supplementary Fig.33). 4) We mentioned another expression of sliding speed in the method: “The load of 2 N was explored; **the reciprocating frequency was 4 Hz**; The lubrication liquid is the saltwater (3.5% of NaCl solution). **The reciprocating stroke is 12 mm along the linear direction of the lower sample.**” This frequency (Hz (cycles/s)) expression is used in standard experiments according to *ASTM G133-22*. We are grateful to the reviewer for reminding us of the sliding speed. We can convert the frequency into an average velocity (48 mm/s) to facilitate the understanding of general readers.

Revised text (main text, page 14, line 297):

By comparison, COF reduced by over 95% between S-PDMS-G/P1-Si/P2-Si and P2-G.

Revised text (main text, page 24, lines 521-524):

The loads of 2, 4, and 6 N were used; the reciprocating frequency was 4 Hz; The lubrication liquid was saltwater (3.5% of NaCl solution). The reciprocating stroke was 12 mm along the linear direction of the lower sample; hence, the average velocity was 48 mm/s.

Revised text (Supplementary Discussion, Section 2.3):

Wear volume generally continues to increase with time. Wear volume is calculated based on the product of sliding distance and wear scar cross-sectional area. The sliding distance follows as:

$$X = 0.002 \times t \times f \times L \quad (S19)$$

$$N = t \times f \quad (S20)$$

where:

X = total sliding distance of the ball, m,

N = number of cycles in the test,

t = test time, s,

f = oscillating frequency, Hz (cycles/s), and

L = length of stroke, mm.

(l) In figure 5D, authors have presented the COF as a function of reciprocal cycles. What is the significance of carrying out this experiment over 12000 cycles. Why the COF values for all cases become different after 4000 cycles. Secondly even in one cycle the normal load and shear load and consequently the COF should exhibit fluctuations. Authors do not make any mention of that. Secondly did the authors obtain shear load at different normal load?

Response:

We appreciate the Reviewer's comments. 1) According to *ASTM G133-22*, the number of cycles is one of the evaluation units in standard friction and wear tests (see the reply to (k)). When carrying out this experiment over 12000 cycles, the stable COF is achieved, allowing us to determine the friction status according to the *Stribeck Curve*. There are many articles that follow the principles (such as *Science Advances*, 2025, 11(4): eadr9834. *Tribology International*, 2023, 189: 108919.). 2) For the question of COF values for all cases becoming different after 4000 cycles, what we understand is the phenomenon that the COF is slightly reduced before 4000 cycles. In fact, all frictions have a running-in process in the early stage. For S-PDMS-G, P1-Si, and P2-Si, the micro-peaks on the contact surfaces are smoothed, and the friction coefficient is slightly reduced. For P1-G and P2-G, the generation of nanoclusters improves surface wettability and thus significantly reduces the COF. 3) The reason why the force and COF within one cycle are not mentioned is that, for practical applications, we are concerned with the average COF over a long period of time. To study normal load (F_z), shear load (F_x), and consequently the COF of one cycle, we captured the cases at cycles

481-484 (before rapid COF reduction) and 12001-12004 (after rapid COF reduction) as shown in Supplementary Fig. 30. All curves of normal load, shear load, and COF show stable periodicity. The normal load (F_z) fluctuates around 2 N. When the applied load is 2 N, F_x exhibits a high value and no sliding plateaus before rapid COF reduction (Supplementary Figs. 30A-D), indicating that that F_x is mainly dominated by the viscoelasticity of the polymer (*Science* 370.6514 (2020): 335-338). While obvious sliding platforms appear after rapid COF reduction (Supplementary Fig. 30E and F), indicating the friction process is caused by the combined action of viscoelasticity and sliding. Subsequently, we compared F_x under 4 cycles under different loads to obtain Supplementary Fig. 31, which showed that the increase in the applied load caused the disappearance of the sliding platform. But the average COF decreases with increasing applied load (Supplementary Fig. 32).

Revised or added Figure:

Supplementary Fig. 30. Load status and the COF of (A) S-PDMS-G and (D) P1-G and in cycles 481-484. Load status and the COF of (B) S-PDMS-G, (C) P1-Si₃N₄, (E) P1-G, and (F) P2-G in cycles 12001-12004.

Supplementary Fig. 31. Changes in shear force under different applied loads

Supplementary Fig. 32. (A) The COF–cycle curve and (B) average COF of P1-G under different loads.

Revised or added text (main text, page 15, line 320):

The coordinated effect of the hydrophilic surface and the porous structure with stored liquid reduced abrasive and three-body wear, causing the boundary lubrication to transform into mixed lubrication, which in turn resulted in ultra-low friction and wear (Fig. 5I). The transformation is reflected by the shear stress and COF under a single reciprocating cycle (Supplementary Fig. 30). Compared to the unreduced state (Supplementary Figs. 30A–D), a clear sliding platform and a stable low COF appeared in the COF-reduced cycle (Figs. 30D and E). At higher loads (4N and 6N), the sliding plateau disappeared, indicating that the friction was dominated by the elastic deformation of the surface⁵⁷; however, the average COF continued to be low (Supplementary Fig. 31).

(m) Page 14 line 309: “...(Figure 22C)...”, There is no such figure!

Response:

We are deeply sorry for missing the word “supplementary”. Due to the addition of a large number of figures, the numbering has changed. We have revised the main text.

Revised or added text (main text, page 14, line 309):

Meanwhile, the wear debris was captured by the pores (Supplementary Fig. 26), leading to the surface wetting change from hydrophobic to hydrophilic (Supplementary Fig. 23B and C), which in turn thickened the lubrication film.

(n) The stretch percentage for some of the samples were found to be quite high, as high as 1250%. What leads to this increase, low crosslinking density, viscoelastic effect, smaller physical entanglement, what? Authors have not thrown any light on the possible mechanism that leads to this increase in extensibility.

Response:

Thank you for the valuable comment. As shown in Supplementary Fig. S43B, HNP-

PDMS@bPdBA cured in -0.1MPa/25 °C shows the lowest tensile modulus. Low entanglement and crosslinking density generally result in low modulus and low elongation (Macromolecules. 2019. 52. 1, 121-134. Macro Materials & Eng. 2021. 306. 12, 2100536). To investigate the cause of the high stretch, we performed dynamic mechanical analysis tests. Loss factor ($\tan\delta$), representing the degree of viscous deformation, can be calculated according to the following formula:

$$\tan\delta = M''/M'$$

where M' and M'' are storage modulus and loss modulus. Storage modulus and loss modulus represent viscosity and elasticity. The $\tan\delta$ of HNP-PDMS@bPdBA cured in -0.1MPa/25°C is much larger than that of HNP-PDMS@bPdBA cured in -0.1MPa/125°C and -0.0MPa/25°C (Supplementary Fig. 44) Therefore, we believe that viscoelastic effect leads to the high stretch of HNP-PDMS@bPdBA cured in -0.1MPa/25°C. The stretch percentage could be improved to 1250%, which is attributed to the increase in the degree of viscous deformation of HNP-PDMS@bPdBA.

Revised or added Figure:

Supplementary Fig. 43. Changes in the $\tan\delta$ of HNP-PDMS@bPdBA cured under different conditions with temperature.

Revised or added text (main text, page 18):

Surprisingly, the stretch percentage could be improved by 1250%, which is attributed to the increase in the degree of viscous deformation of HNP-PDMS@bPdBA, as shown in Supplementary Fig. 44 (see Supplementary Information, Section 2.5, for more details). Additionally, the performance of HNP-PDMS@bPdBA was better than that of P-PDMS@Sugar (Fig. 6G).

Revised or added text (Supplementary Discussion Section 2.4):

As shown in Supplementary Fig. S43B, HNP-PDMS@bPdBA cured in -0.1MPa/25 °C shows lowest tensile modulus. Low entanglement and crosslinking density generally result in low modulus and low elongation^{8,9}. To investigate the cause of the high stretch, we performed dynamic mechanical analysis tests. Loss factor ($\tan\delta$), representing the

degree of viscous deformation, can be calculated according to the following formula:

$$\tan\delta = M''/M' \quad (\text{S21})$$

where M' and M'' are storage modulus and loss modulus. Storage modulus and loss modulus represent viscosity and elasticity. The $\tan\delta$ of HNP-PDMS@bPdBA cured in -0.1MPa/25°C is much larger than that of HNP-PDMS@bPdBA cured in -0.1MPa/125°C and -0.0MPa/25°C (Supplementary Fig. 44) Therefore, we believe that viscoelastic effect leads to the high stretch of HNP-PDMS@bPdBA cured in -0.1MPa/25°C. The stretch percentage could be improved to 1250%, which is attributed to the increase in the degree of viscous deformation of HNP-PDMS@bPdBA.

Revised or added text (method, Characterizations):

Dynamic mechanical analysis (DMA) was performed using a DMA equipment (Netzsch, Germany, DMA242) with a heating rate of 3 °C/min from -20 °C to 100 °C.

Reviewer #3 (Remarks to the Author):

The authors report a novel, template-free emulsion method to create purely porous silicone elastomers by driving heterogeneous network formation using borate-based dynamic crosslinkers. By combining a borate-modified PDMS precursor with a commercial PDMS prepolymer and inducing spontaneous phase separation under controlled thermal and vacuum conditions, they generate an interconnected porous structure after selectively removing the sacrificial borate phase. The resulting materials exhibit remarkably low friction and wear under aqueous lubrication while retaining excellent stretchability and toughness. Furthermore, the approach allows precise tuning of pore morphology and mechanical properties through processing parameters and can be extended to other silicone elastomer systems, demonstrating broad potential for applications in low-friction, wear-resistant soft materials. However, the manuscript, in its current state, may not meet the standards for publication in Nature Communications, a journal known for its groundbreaking and influential contributions to the scientific community. The idea and concept of the paper are excellent and novel, but the available data on this particular aspect appears to be insufficient. Furthermore, there is a need for more comprehensive supporting evidence and data to bolster the proposed hypotheses. The authors are encouraged to consider the following suggestions for enhancing the manuscript in preparation for future submissions.

To strengthen the manuscript, the authors should consider (Major):

1. To fulfill the objectives proposed by the authors, additional stability and wear-rate measurements under thermal, high-temperature, UV, and aging conditions must be conducted.

Response:

We appreciate the Reviewer's comments. We added additional stability tests, such as friction under UV light and at 60 °C (which is high temperature relative to water), salt water immersion for 3 weeks, and cyclic stretching for 2000 times to treat P1. Besides, we performed 122400 cycles (test times: 8.5 h) and calculated the wear rate. As shown in Supplementary Fig. 35, regardless of the method used to treat the surface, a sharp decrease in the COF always occurs. The average COF remains 0.03 ~ 0.2. An increase in the COF at 60 °C after 10000 cycles, which is due to the evaporation of water. However, the low COF state can be restored after adding salt water (see next response). In any case, the wear rate of the aging sample ($< 6.69 \times 10^{-6} \text{ mm}^3/\text{N}\cdot\text{m}$) was reduced by over 75% compared with the wear rate of S-PDMS-G, as shown in Supplementary Fig. 36. In particular, when the friction time was extended to 8.5 h (122400 cycles), the wear rate was further reduced to as low as $\sim 8.62 \times 10^{-6} \text{ mm}^3/\text{N}\cdot\text{m}$.

Revised or added Figure:

Supplementary Fig. 35. (A) The COF–cycle curve and (B) average COF of P1-G after aging or degradation treatment.

Supplementary Fig. 36. The wear rate of P1-G and its after aging or degradation treatment.

2. Beyond the current stability tests over several thousand cycles, experiments addressing a much longer fatigue life are required. Considering the applications the authors propose, stability over only a few thousand cycles is insufficient.

Response:

Thank you for the significant comment. We performed the long-term wear test ($>1.2 \times 10^5$ cycles) for P1-G. As reciprocating cycles progress and water evaporates, causing the COF to increase, the COF quickly decreases to a lower level after adding a little water (~ 0.1 mL) as shown in Supplementary Fig. 33.

Revised or added Figure:

Supplementary Fig. 33. The COF–cycle curve of P1-G after 122400 reciprocating cycles (Yellow area: Lubrication deteriorated as the water evaporated; Blue area: Lubrication improved after adding 0.1 mL of salt water).

Revised or added text (Supplemental Information, Section 2.3):

To investigate the stability of the low-friction transition and low-wear state, we perform fatigue and aging tests on the P1 (HNP-PDMS@BA-125/-0.1MPa). As shown in Supplementary Fig. 33, with reciprocating cycles progressing and water evaporating, the COF increases, but the COF quickly decreases to a lower level after adding a little (~0.1 mL) salt water. SEM and EDS results show that the pores of P1 are always able to capture the wear debris during this process (Supplementary Fig. 34A). COF results of other tests are shown in Supplementary Fig. 35. Friction testing of the sample after cyclic stretching (over 2000 cycles, with a stretch ratio of 20% in each cycle) revealed that it still exhibited low friction, with the average COF less than 0.15. Furthermore, after 3 weeks of salt water immersion, UV irradiation (365 nm), and heating treatment (60 °C), P1 can stably reduce the COF. An increase in the COF at 60 °C after 10000 cycles, which is due to the evaporation of water. However, the low COF state can be recovered after adding salt water. In any case, the wear rate of the aging sample ($< 6.69 \times 10^{-5} \text{ mm}^3/\text{N}\cdot\text{m}$) was reduced by over 75% compared with the wear rate of S-PDMS-G, as shown in Supplementary Fig. 36. In particular, when the friction time was extended to 8.5 h (122400 cycles), the wear rate was further reduced to as low as $\sim 8.62 \times 10^{-6} \text{ mm}^3/\text{N}\cdot\text{m}$. We repeated the friction test of P1-G three times in situ without processing and found that COF reduction still occurred (Supplementary Fig. 37). After repeated friction tests, the CA is still close to that after the first friction test (Supplementary Figs. 23C and D). The corresponding SEM and EDS results of the worn surface are shown in Supplementary Fig. 34B, which reveals the presence of wear debris and iron in the pores, indicating that the nanoclusters were still trapped. To study the effect of product detachment on friction, after the first friction test, the mixed liquid on the surface was wiped off by the dust-free fabric and the new friction test was carried out. The COF dropped rapidly with almost no

running-in period of thousands of cycles (Supplementary Fig.38). Compared to S-PDMS-G, P1-Si, and P2-Si, the COF of P1-Si still dropped by over 60%. We infer that the old ball with products still works. To verify that, we changed the new ball after the mixed liquid on the surface was wiped off by the dust-free fabric and performed a repeated friction test. The friction results show that a running-in period of thousands of cycles is necessary under this condition (Supplementary Fig.39). Overall, all these indicate that this porous surface has the potential for long-term service.

3. The authors have validated their method in PDMS and Ecoflex. The claim that the fabrication approach applies to all polysiloxane systems is unsupported by evidence from only two examples; additional validation cases are needed to substantiate this assertion.

Response:

Thank you for pointing out the potential issue, and this is an important comment. To address the issue, we added three other elastomer materials (DC170, DC527, and RTV615) for validation. Using the same preparation method as HNP-PDMS, we successfully prepared new porous elastomers (DC170@BA, DC527@BA, and RTV615@BA) as shown in Supplementary Fig. 46. These porous elastomers have the potential to improve the elongation.

Revised or added Figure:

Supplementary Fig. 46. The surface SEM images of (A) DC170, (B) DC170@BA, (D) DC527, (E) DC527@BA, (G) RTV615, and (H) RTV615@BA. Stress-stretch curves of (C) DC170 and DC170@BA, (F) DC527 and DC527@BA, (J)RTV615 and RTV615@BA.

Revised text (main text, Section 2.6):

The introduction of D-PDMS@BA significantly delayed the fracture of the elastomer, and the toughness increased to 150 %, despite the elastomer being porous. Furthermore, we successfully prepared new porous elastomers using three other PDMS materials (DC170, DC527, and RTV615), as shown in Supplementary Fig. 46. These porous elastomers also exhibited elongation improvement potential. These results proved that our method was applicable to the preparation of mechanically enhanced porous elastomers for all PDMS systems.

4. The authors' scientific novelty and extensive material analyses, including modeling, are highly commendable; demonstrating a practical application is necessary to better align with the scope of Nature Communications. Any example—such as the machinery, robots, bearings, or biological models—would be suitable.

Response:

We are very grateful for your recognition and comments. In the previous manuscript, we mentioned the potential application of our material in flexible sensors. In this revised process, we performed a set of experiments to demonstrate the practical application. To prepare flexible sensors, conductive properties need to be given to PDMS materials due to the insulating nature of PDMS. We used carbon fibers with a diameter of 7 μm and a length of 33.6 μm as the conductive material (Supplementary Fig. 47). It is difficult to form a pathway by directly mixing PDMS with conductive materials (Supplementary Fig. 48A). Based on HNP-PDMS@BA, however, we added carbon fibers and successfully prepared conductive silicone (Resistance < 1k Ω) without the need for additional surfactants (Supplementary Figs. 48B and C). Besides, the mechanical performance of HNP-PDMS@C/-0.1MPa is better than that of S-PDMS (Figure 6J). We attribute the improved performance of HNP-PDMS@C/-0.1MPa to the dispersion of carbon fibers within the through-pores of HNP-PDMS (Fig. 6K). This flexible conductive material can be used as the skin of the bionic robot hand to sense force and identify materials (Fig. 6L). To demonstrate the application effect, HNP-PDMS@C/-0.1MPa is encapsulated in PDMS and then bent and fixed on the robotic finger to contact the wall Supplementary Fig. 49A). In this case, our material can act as both a strain sensor and a triboelectric sensor. As the contact force increases from 1N to 4N, the change rate of resistance also improves accordingly (Supplementary Fig. 49B). According to the peak value and width of the wave, various materials can be distinguished as shown in Supplementary Fig. 49C (ACS Appl. Mater. Interfaces. 2024. 16. 43, 58980-58990). These demonstrate practical potential applications.

Revised or added Figure:

Supplementary Fig. 47. (A) Single carbon fiber (B) Distribution of carbon fiber length.

Supplementary Fig. 48. Resistance of PDMS/C, HNP-PDMS@C/-0.0, and HNP-PDMS@C/-0.1 (Sample size: 10 mm×10 mm×1 mm).

Fig. 6. (J) Stress-stretch curves of PDMS, HNP-PDMS@C/-0.0, and HNP-PDMS@C/-0.1. (K) Cross-sectional SEM image of HNP-PDMS@C/-0.1. (L) Contact sensing diagram.

Supplementary Fig. 49. (A) Actual picture of the bionic robot hand contacting the material. (B) Resistance changes under different forces (R_0 and R are the initial resistance and the resistance under pressure, contact speed: 50 mm/min). (C) Output voltage (U) curves of 6 different materials under the same conditions (Contact speed: 500 mm/min).

Revised or added text (main text, Section 2.6):

To prepare flexible sensors, conductive properties need to be introduced to PDMS materials, as PDMS is inherently insulating. We used carbon fibers with a diameter of 7 μm and a length of 33.6 μm as the conductive material (Supplementary Fig. 47). It is difficult to form a pathway by directly mixing PDMS with conductive materials (Supplementary Fig. 48A). To achieve conductivity in PDMS/silicone materials, the conductive material is usually connected within the polymer using a similar preparation method as porous materials.^{58,59} Based on the formation mechanism of HNP-PDMS@BA, we added carbon fibers and successfully prepared conductive PDMS (Resistance < 1k Ω), again without the need for additional surfactants or emulsions (Supplementary Figs. 48B and C). Besides, the mechanical performance of HNP-PDMS@C/-0.1MPa is better than that of S-PDMS (Figure 6J). We attribute the improved performance of HNP-PDMS@C/-0.1MPa to the dispersion of carbon fibers within the through-pores of HNP-PDMS (Figure 6K). This flexible conductive material can be used as the skin of the bionic robot hand to sense force and identify materials (Figure 6L). To demonstrate the application effect, HNP-PDMS@C/-0.1MPa is encapsulated in PDMS and then bent and fixed on the robotic finger to contact the wall

(Supplementary Fig. 49A). In this case, our material can act as both a strain sensor and a triboelectric sensor. As the contact force increases from 1N to 4N, the change rate of resistance also improves accordingly (Supplementary Fig. 49B). According to the peak value and width of the wave³⁹, various materials can be distinguished as shown in Supplementary Fig. 49C, demonstrating practical potential applications.

Revised or added text (main text, method):

Fabrication of Flexible Sensor Material

The preparation method of HNP-PDMS@C differs from that of HNP-PDMS@BA; the carbon fiber is added (25% of the total mass) during the mixing of the S-PDMS and the D-PDMS@BA prepolymers. Finally, the unstable D-PDMS@BA network was removed to obtain the HNP-PDMS@C by ultrasonic cleaning of the composite elastomer with ethanol. Two 1-mm thick S-PDMS sheets sandwiched a 1-mm thick HNP-PDMS@C to create the flexible sensor material.

Characterizations

Dynamic mechanical analysis (DMA) was performed using a DMA equipment (Netzsch, Germany, DMA242) with a heating rate of 3 °C/min from -20 °C to 100 °C. An oscilloscope (Tektronix TBS2072B) was employed to capture and record the open circuit voltage. a LabVIEW-controlled digital source meter (Keithley 2400, America) recorded the resistance in real time.

Response Letter

We are grateful to the editor and reviewers for their comments on our article. [Referees' comments are in black; Author responses are in blue]

Reviewer #1 (Remarks to the Author):

The authors have fully addressed my concerns. I suggest the publication of the revised manuscript.

Response:

We appreciate the reviewer for the positive comments and recognition of our work. The manuscript has greatly benefited from them.

Reviewer #2 (Remarks to the Author):

[In correspondence with the editors, the referee expressed that the manuscript is ready for publication as-is.]

Response:

We are very grateful to the reviewer for the comments and recognition of our work. The manuscript has greatly benefited from them.

Reviewer #3 (Remarks to the Author):

The authors present their research in a logical and coherent manner, supported by robust experimental design, sound methodology, and convincing data analysis. The novelty of the work is evident in the way it bridges current knowledge gaps and provides new insights that are likely to stimulate further research and practical applications. I find no substantive weaknesses that would require revision, and I am pleased to recommend the manuscript for acceptance in its current form.

Response:

We appreciate the reviewer for the positive comments and recognition of our work. The manuscript has been greatly improved.